# Small Updates, Big Doubts:
# Does Parameter-Efficient Fine-tuning Enhance Uncertainty Awareness for Large Language Models?

## Abstract

Parameter-efficient fine-tuning (PEFT) is the de facto approach for adapting large language models (LLMs), yet its effect on model uncertainty—the very signal that hallucination detectors rely on—remains poorly understood. We present a controlled empirical study of how PEFT reshapes uncertainty-aware hallucination detection in fact-seeking question answering. Across three open-weight LLMs (LLaMA-3.2-3B, Qwen-2.5-3B, Mistral-7B), three QA benchmarks (TriviaQA, NQ-Open, SQuAD), and seven answer-level detectors spanning semantic-consistency, confidence, and entropy families, we find a striking asymmetry: PEFT improves answer accuracy by only 1–3% on average, yet boosts detection AUROC by up to 8.7%, with consistent gains across most black-box detectors. Behavioral analysis reveals that this improvement stems largely from PEFT shifting uncertainty scores away from overconfident error regimes, with gains from improved factual knowledge appearing to play a more limited role. In contrast, white-box linear probes on hidden states show inconsistent results, indicating that PEFT reshapes how uncertainty is expressed more than how correctness is linearly encoded. Our findings demonstrate that, in fact-seeking QA, PEFT acts primarily as an uncertainty reshaper that makes incorrect answers more detectable, and we caution that these results concern answer-level detection with external verification rather than open-ended generation.

## 1 Introduction

Hallucination, the generation of fluent but fabricated or factually incorrect content, remains a central obstacle to deploying LLMs in knowledge-intensive applications (Du et al., 2024; Manakul et al., 2023; Qiu & Miikkulainen, 2024). Although often discussed as a factual failure, it also has an epistemic dimension: models can be confidently wrong (Tian et al., 2024; Kalai & Vempala, 2024). In practice, safety improves not only by reducing incorrect answers, but also by ensuring that incorrect answers are accompanied by uncertainty signals that support abstention, fallback, escalation, or filtering. From this perspective, systems benefit not only when errors become rarer, but also when they become more detectable from model behavior (Wang et al., 2024; Li et al., 2024; Lu et al., 2024).

At the same time, LLMs are routinely adapted using PEFT methods such as LoRA (Hu et al., 2022), DoRA (Liu et al., 2024), and PiSSA (Zhu et al., 2024). PEFT can approach full fine-tuning performance while updating only a small fraction of parameters, and has become a de facto standard in practice. Yet PEFT is typically evaluated through downstream accuracy or overall hallucination rate, leaving its effect on answer-level error detectability underexplored. This gap is important because widely used uncertainty-based detectors rely on behavioral signals, such as entropy, self-consistency, and semantic agreement, that PEFT may substantially reshape.

Fact-seeking open-ended QA benchmarks provide a natural setting for studying this question because they combine (i) generation freedom, (ii) factual grounding, and (iii) retrieval of knowledge from model parameters rather than closed-set classification. At the same time, this setting allows externally verifiable answer

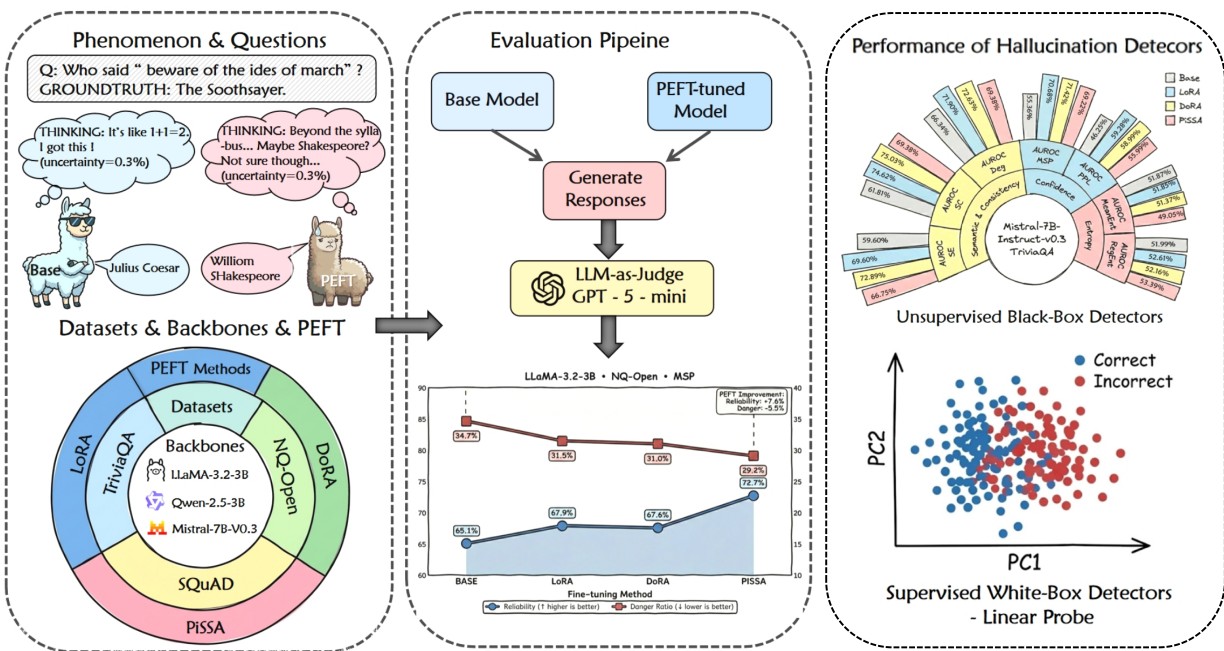

Figure 1: The overview of out empirical study of how hallucination detection ability is affeted by parameter-efficient fine-tuning.

correctness, making it possible to distinguish ground-truth answer errors from the detector scores used to identify them. In this paper, we focus on *answer-level hallucination detection* in fact-seeking QA, rather than claim-level hallucination in open-ended long-form generation. Accordingly, our goal is not to argue that PEFT universally improves hallucination behavior across tasks, but to study whether PEFT changes the uncertainty signals that make externally verified incorrect answers more or less detectable in this controlled setting.

We address this question through a systematic empirical study of PEFT under canonical answer-level hallucination detection settings. We compare LoRA, DoRA, and PiSSA across three open-weight backbones (LLaMA, Mistral, and Qwen) and three fact-seeking QA benchmarks (TriviaQA, NQ-Open, and SQuAD). Beyond QA accuracy, we evaluate a diverse suite of answer-level uncertainty detectors spanning semantic-consistency-based, confidence-based, and entropy-based signals, and we further analyze hidden-state changes using linear probing and PCA.

Our study yields three main conclusions:

- **Conclusion #1: Accuracy changes are modest.** PEFT yields only marginal gains in QA accuracy, suggesting limited direct reduction of answer errors through improved factual correctness alone.

- **Conclusion #2: Incorrect answers become more detectable under several answer-level detectors.** Despite modest accuracy improvements, PEFT consistently improves the separability of externally verified incorrect answers from correct ones for a broad set of black-box detectors, especially semantic-consistency-based and confidence-based methods. Our empirical results suggest that one important factor is a shift away from highly overconfident error regimes.

- **Conclusion #3: White-box probing results are mixed.** PEFT induces structured shifts in hidden representations related to uncertainty, but it does not uniformly increase the linear separability of correctness labels. As a result, supervised white-box linear-probe performance is less consistent than the improvements observed for black-box detector signals.

Overall, our findings suggest that, in fact-seeking QA, PEFT primarily reshapes detector-relevant uncertainty signals rather than substantially expanding factual knowledge. In this sense, PEFT appears to improve the detectability of externally verified answer errors under several commonly used answer-level detectors. We therefore interpret our results as evidence about uncertainty signal reshaping and calibration-like score separability in this setting, rather than as a general claim that PEFT improves hallucination behavior across all generation tasks. To our knowledge, this is the first unified study that examines PEFT from this perspective by jointly evaluating external answer-level detectors and internal representation probes in a controlled QA setting.

## 2 Related Work

**Uncertainty Estimation:** Fact-seeking QA is a standard evaluation setting for answer reliability and hallucination-related errors, with benchmarks such as SQuAD, TriviaQA, and Natural Questions providing reference answers (Kwiatkowski et al., 2019). A major line of prior work estimates epistemic uncertainty from *sample-based* consistency and semantic agreement. SelfCheckGPT compares a model's answer against self-generated alternatives (Manakul et al., 2023), while Semantic Entropy aggregates semantic dispersion among sampled answers (Farquhar et al., 2024). Follow-up work further proposes semantic entropy probes that map hidden representations to semantic clusters for cheaper detection (Kossen et al., 2024). Complementary logit-based signals adapt maximum softmax probability and predictive entropy from out-of-distribution detection (Hendrycks & Gimpel, 2017; Farquhar et al., 2024), although calibration remains challenging in LLMs (Kadavath et al., 2022; Lin et al., 2022). Despite rapid progress in uncertainty estimation and answer-level detectors, there is still limited systematic understanding of how fine-tuning, and PEFT in particular, changes the uncertainty and consistency signals on which these methods rely.

**Hallucination Detection:** Prior work on hallucination detection can be broadly grouped into two settings. The first studies *answer-level hallucination detection* in question answering, where the goal is to determine whether a model's answer to a question is correct or incorrect, typically using fact-seeking benchmarks with reference answers. This line includes methods based on self-consistency, semantic agreement, confidence, and entropy signals, which infer answer reliability from the model's own outputs or predictive uncertainty. The second studies *claim-level hallucinated fact detection*, which focuses on identifying fabricated or unsupported factual claims within generated text, often at the sentence or claim level in long-form generation, summarization, or dialogue. Compared with QA-based answer-level detection, this setting usually requires finer-grained claim verification, often with external evidence or retrieval. Our work belongs to the first category. Specifically, we study answer-level hallucination detection on fact-seeking QA tasks and investigate how PEFT affects the detector-relevant uncertainty signals used in this setting, rather than claim-level verification of individual facts in free-form text. Accordingly, our contribution is not to redefine hallucination itself, but to analyze how PEFT changes the detectability of externally verified answer errors under commonly used answer-level detectors.

**Parameter-Efficient Fine-Tuning and Knowledge Dynamics:** PEFT adapts LLMs by training a small set of additional parameters while often approaching full fine-tuning performance. We study three widely used low-rank methods, LoRA (Hu et al., 2022), PiSSA (Zhu et al., 2024), and DoRA (Liu et al., 2024), which share a common low-rank adaptation core but differ in their parameterization. LoRA adds trainable low-rank adapters to frozen weights (Hu et al., 2022), PiSSA uses SVD-based initialization from the original weights (Zhu et al., 2024), and DoRA decomposes weights into magnitude and direction and adapts directions separately (Liu et al., 2024). Beyond accuracy, fine-tuning can reorganize knowledge access rather than simply inject facts: narrow supervision may increase related-query hallucinations (Gekhman et al., 2024), and knowledge editing can struggle with global consistency (Zhang et al., 2024). It may also improve truthfulness while yielding mixed calibration (Tian et al., 2024), and instruction tuning can reduce blatant errors yet increase overconfidence on remaining mistakes (Perez et al., 2022). Motivated by these observations, we evaluate PEFT through the lens of answer-level error detectability and uncertainty signal reshaping, focusing on how PEFT changes detector-relevant behavior in fact-seeking QA rather than making a general claim about hallucination behavior across all generation settings.

## 3 Experiment

### 3.1 Experimental Setup

**Backbones and datasets.** We consider three open-weight instruct models: LLaMA-3.2-3B-InstructGrattafiori et al. (2024), Qwen2.5-3B-InstructYang et al. (2025), and Mistral-7B-Instruct-v0.3Jiang et al. (2023). For each backbone, we compare the base instruct model against its PEFT variants (LoRA, PiSSA, DoRA), fine-tuned in-domain on each dataset. Fact-seeking benchmarks are typically presented as QA tasks with short target spans. Our goal is to isolate how PEFT affects hallucination detectors. To make this analysis meaningful, we evaluate on fact-seeking QA benchmarks where these detectors are commonly studied and known to behave reliably: TriviaQA (`rc.nocontext`)[1], Natural Questions (NQ-Open)[2], and SQuAD v1[3] to probe hallucination behavior. The data are attached in appendix A.

**Parameter-efficient fine-tuning methods.** We fine-tune all models using the Hugging Face `peft` library. Use rank $r = 32$, scaling factor $\alpha = 64$, and dropout $p = 0.05$, AdamW with learning rate $2 \times 10^{-5}$, global batch size 64, warmup ratio 0.03, and train for 1 epoch in bfloat16. And target the attention projection layers (`q_proj`, `k_proj`, `v_proj`, `o_proj`) for all PEFT methods. Training and evaluation loss curves for all methods and datasets are provided in Appendix C (Figures 6 and 7).

**Black-box hallucination detectors.** We evaluate task accuracy alongside seven hallucination detectors, which fall into three groups. **Semantic consistency based methods** exploit multiple stochastic samples: Semantic Entropy (SE)Farquhar et al. (2024) measures entropy over semantically clustered responses, SelfCheckGPT (SC)Manakul et al. (2023) counts how many samples support the primary answer, and Degree of Uncertainty (Deg)Lin et al. (2023) measures connectivity between sampled responses. **Confidence based methods** Maximum sequence probability score leverages the probability of the most likely sequence generationFadeeva et al. (2023):

$$\text{MSP}(\mathbf{y} \mid \mathbf{x}, \theta) = 1 - P(\mathbf{y} \mid \mathbf{x}, \theta).$$

Perplexity is computed as

$$P(\mathbf{y}, \mathbf{x}; \theta) = \exp\left\{ -\frac{1}{L} \log P(\mathbf{y} \mid \mathbf{x}, \theta) \right\}$$

**Entropy based methods** measure predictive dispersion: Predictive EntropyFarquhar et al. (2024), also called Monte Carlo Sequence Entropy, is the negative average of log probability of response. Mean Token EntropyFadeeva et al. (2023) averages token entropy across the sequence.

**White-box hallucination detector: linear probe.** Following Du et al. (2024); Kadavath et al. (2022); Kossen et al. (2024), we adopt the standard approach of training an external classifier for hallucination prediction using the backbone LLM's hidden states as input. Concretely, we train logistic-regression models on a labeled training set, where labels indicate response correctness. We split the original validation set into two equal halves, which we use as a new validation set and a held-out test set, respectively. We select the hidden-state layer used as classifier input by maximizing performance on the new validation set.

**Evaluation metrics.** Following Yao et al. (2025), We adopt LLM-as-judge method using GPT-5-mini to label the generated response. And We report AUROC (Area Under the Receiver Operating Characteristic Curve) as the primary metric for hallucination detection. For SQuAD, we additionally emphasize AUPR (Area Under Precision-Recall Curve) due to severe class imbalance: as an extractive reading comprehension task where answers are explicitly present in the context, SQuAD yields high model accuracy (>90%) and consequently low hallucination rates (<10%). Under such imbalance, AUPR provides a more informative assessment of detection performance on the minority class (hallucinations) than AUROC (Davis & Goodrich, 2006). To validate the reliability of this protocol, we also report a cross-judge consistency analysis in Appendix B. Tables 7 and 8 show highly consistent verdicts across judges, with pairwise agreement above 97%.

---

[1] https://huggingface.co/datasets/mandarjoshi/trivia_qa
[2] https://github.com/google-research-datasets/natural-questions
[3] https://huggingface.co/datasets/rajpurkar/squad

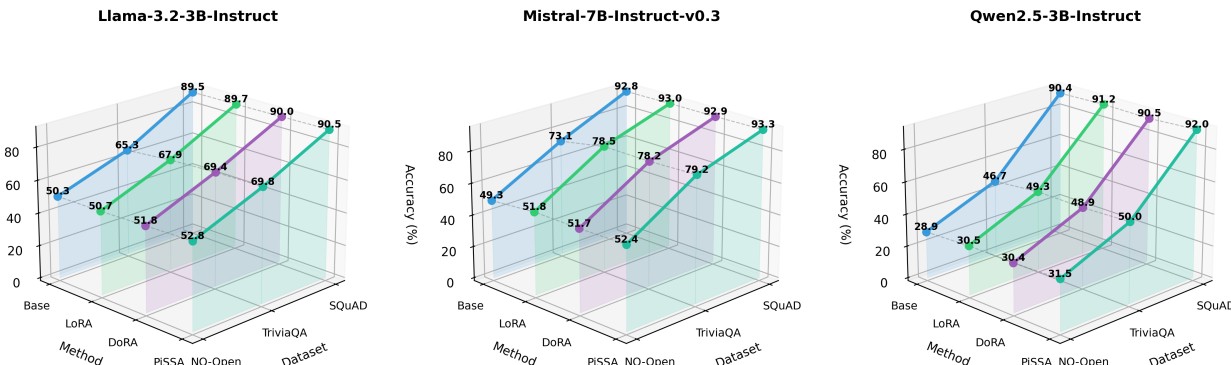

Figure 2: Test accuracy across three backbones, three datasets, and four methods. Each panel shows a 3D waterfall visualization where the x-axis shows datasets, y-axis shows methods, and z-axis shows test accuracy (%). In this paper, we define the marginal when the changes are within 1%

## 3.2 Experimental Results

Table 1: AUROC scores for hallucination detection baselines on LLaMA-3.2-3B-Instruct. The highest metric in each configuration is bolded. Compared to the small accuracy changes in Figure 2, the effects on these hallucination detection baselines are much more pronounced and structured. The improvement equals the best highlighted scores minus base scores. And the average improvement represents the average improvement on this dataset with the corresponding hallucination detector. The after tables follow these colour highlighting setting.

| Dataset | Method | Semantic Consistency | | | Confidence | | Entropy | |
|---|---|---|---|---|---|---|---|---|
| | | SE | SC | Deg | MSP | Perplexity | Mean Ent. | Pre Ent. |
| NQ-Open | Base | 0.7028 | 0.7432 | 0.7376 | 0.6977 | 0.7263 | 0.7295 | 0.6649 |
| | DoRA | 0.7453 | 0.7599 | 0.7379 | 0.7585 | 0.7445 | 0.7163 | 0.7030 |
| | LoRA | 0.7451 | 0.7538 | 0.7416 | 0.7620 | 0.7507 | 0.7233 | 0.7058 |
| | PiSSA | **0.7601** | **0.7806** | **0.7615** | **0.7773** | **0.7648** | **0.7381** | **0.7309** |
| | ▲Best Impr. | +5.73% | +3.74% | +2.39% | +7.96% | +3.85% | +0.86% | +6.60% |
| | ▲Avg Impr. | +4.74% | +2.16% | +0.94% | +6.82% | +2.70% | -0.36% | +4.83% |
| TriviaQA | Base | 0.8205 | 0.8737 | 0.8703 | 0.8154 | 0.8328 | **0.8457** | 0.7819 |
| | DoRA | **0.8823** | 0.8838 | 0.8738 | 0.8866 | **0.8605** | 0.8202 | 0.8138 |
| | LoRA | 0.8737 | 0.8889 | **0.8779** | 0.8656 | 0.8597 | 0.8303 | **0.8216** |
| | PiSSA | 0.8797 | **0.8896** | 0.8772 | **0.8878** | 0.8595 | 0.8275 | 0.8087 |
| | ▲Best Impr. | +5.92% | +1.59% | +0.76% | +7.24% | +2.77% | -1.54% | +3.97% |
| | ▲Avg Impr. | +5.81% | +1.37% | +0.60% | +6.46% | +2.71% | -1.97% | +3.28% |
| SQuAD | Base | 0.7158 | 0.6978 | 0.6764 | 0.6972 | 0.6429 | 0.6381 | **0.6808** |
| | DoRA | 0.7992 | **0.7715** | 0.6801 | 0.6974 | 0.6901 | 0.6923 | 0.6336 |
| | LoRA | **0.8027** | 0.7130 | **0.7313** | **0.7346** | **0.7074** | **0.7006** | 0.6476 |
| | PiSSA | 0.7775 | 0.7107 | 0.6855 | 0.6988 | 0.6834 | 0.6863 | 0.6107 |
| | ▲Best Impr. | +8.69% | +7.37% | +5.49% | +5.54% | +6.45% | 6.25% | -3.30% |
| | ▲Avg Impr. | +7.73% | +3.39% | +2.26% | +1.31% | +5.07% | +5.50% | -5.02% |

Table 2: AUROC scores for hallucination detection baselines on Qwen2.5-3B-Instruct.

| Dataset | Method | Semantic Consistency | | | Confidence | | Entropy | |
|---|---|---|---|---|---|---|---|---|
| | | SE | SC | Deg | MSP | Perplexity | Mean Ent. | Pre Ent. |
| NQ-Open | Base | 0.7666 | 0.7761 | 0.7912 | 0.7193 | 0.7173 | **0.7225** | 0.6920 |
| | DoRA | **0.7780** | **0.8233** | 0.7990 | 0.7883 | 0.7264 | 0.6843 | 0.7018 |
| | LoRA | 0.7723 | 0.8189 | 0.7936 | 0.7817 | 0.7219 | 0.6800 | 0.7041 |
| | PiSSA | 0.7765 | 0.8178 | **0.7995** | **0.7946** | **0.7297** | 0.6874 | **0.7116** |
| | ▲Best Impr. | +1.14% | +5.17% | +0.83% | +7.53% | +1.24% | -3.51% | +1.96% |
| | ▲Avg Impr. | +0.90% | +4.39% | +0.62% | +6.89% | +0.87% | -3.86% | +1.38% |
| TriviaQA | Base | 0.8318 | 0.8551 | 0.8548 | 0.8132 | 0.8155 | **0.8320** | 0.7808 |
| | DoRA | **0.8707** | 0.8756 | 0.8642 | 0.8938 | 0.8508 | 0.8076 | 0.7899 |
| | LoRA | 0.8694 | 0.8796 | 0.8647 | 0.8971 | **0.8549** | 0.8145 | **0.7918** |
| | PiSSA | 0.8692 | **0.8828** | **0.8710** | **0.8973** | 0.8497 | 0.8114 | 0.7898 |
| | ▲Best Impr. | +3.89% | +2.77% | +1.62% | +8.41% | +3.94% | -1.75% | +1.10% |
| | ▲Avg Impr. | +3.80% | +2.42% | +1.18% | +8.29% | +3.63% | -2.08% | +0.97% |
| SQuAD | Base | 0.6669 | 0.6109 | 0.6599 | 0.6683 | 0.6519 | 0.6558 | 0.6314 |
| | DoRA | 0.7629 | 0.7412 | 0.7782 | 0.7552 | 0.7303 | 0.7313 | 0.6266 |
| | LoRA | 0.7533 | 0.7566 | **0.7810** | 0.7436 | 0.7295 | 0.7410 | **0.6488** |
| | PiSSA | **0.7959** | **0.7608** | 0.7728 | **0.7596** | **0.7460** | **0.7570** | 0.6246 |
| | ▲Best Impr. | +12.90% | +14.99% | +12.11% | +9.13% | +9.41% | +10.12% | +1.74% |
| | ▲Avg Impr. | +10.38% | +14.20% | +11.74% | +8.45% | +8.34% | +8.73% | +0.19% |

💡 **Takeaway #1: PEFT yields modest hallucination mitigation but significant hallucination detection improvement on QA datasets.** Figure 2 visualizes the test set accuracy of QA datasets across models and datasets. In NQ-Open, where base accuracy is overall low, LoRA, PiSSA, and DoRA lift performance by only 0.4% to 3.1%, around 1.5% hallucination mitigation on average. In SQuAD, base models already approach the performance ceiling ($\approx$90%), and PEFT yields negligible gains of 0.1% to 1.6%. In TriviaQA, improvements range from 2.6% to 6.1%, about 3% hallucination mitigation on average. Across all configurations, PiSSA tends to outperform other methods obviously, consistent with it being the most recent advancement in PEFT.

💡 **Takeaway#2: Semantic consistency based and confidence based hallucination detectors are improved significantly but entropy based detectors have marginal improvement after PEFT.** Tables 1, 2 and 3 report AUROC scores for all hallucination detection baselines across three backbones and three QA datasets. Across all backbones and datasets, semantic consistency based and confidence based detectors consistently improve after PEFT based AUROC.

Comparing the based and fine-tuning models across all datasets, PEFT consistently improves semantic consistency based and confidence based methods across all backbones and datasets. **From the average improvement of AUROC**, the semantic and consistency based hallucination detectors and confidence based methods systematically raise by ranging from 0.94% (NQ-Open with degree of uncertainty) to 7.73% (SQuAD with semantic entropy) on LLaMA-3.2-3B. And for Qwen-2.5-3B, the performance has been uplifted by from 0.62% (NQ-Open with degree of uncertainty) to 14.20% (SQuAD with SelfCheckGPT). As for Mistral-7B, the scores are increased by 1.70% (NQ-Open with degree of uncertainty) to 11.18% (TriviaQA with MSP).

In contrast to semantic consistency-based and confidence-based hallucination detectors, entropy-based detectors exhibit inconsistent performance. We hypothesize that token-level entropy measures capture lexical diversity and syntactic variation, which are influenced by factors orthogonal to factual correctness, such as vocabulary choice, sentence structure, and generation randomness. In contrast, semantic consistency-based

Table 3: AUROC scores for hallucination detection baselines on Mistral-7B-Instruct-v0.3.

| Dataset | Method | Semantic Consistency | | | Confidence | | Entropy | |
| | | SE | SC | Deg | MSP | Perplexity | Mean Ent. | Pre Ent. |
|---|---|---|---|---|---|---|---|---|
| NQ-Open | Base | 0.7047 | 0.7357 | 0.7578 | 0.6791 | 0.6499 | 0.6680 | 0.6658 |
| | DoRA | 0.7690 | 0.7875 | 0.7730 | 0.7826 | 0.7271 | 0.6953 | 0.7123 |
| | LoRA | 0.7684 | 0.7922 | **0.7769** | **0.7894** | **0.7368** | **0.7058** | **0.7267** |
| | PiSSA | **0.7712** | **0.7937** | 0.7746 | 0.7850 | 0.7318 | 0.7038 | 0.7193 |
| | ▲Best Impr. | +6.65% | +5.80% | +1.91% | +11.03% | +8.69% | +3.78% | +6.09% |
| | ▲Avg Impr. | +6.48% | +5.54% | +1.70% | +10.66% | +8.20% | +3.36% | +5.36% |
| TriviaQA | Base | 0.7869 | 0.7966 | 0.8243 | 0.7571 | 0.7185 | 0.7396 | 0.7448 |
| | DoRA | **0.8891** | **0.9017** | **0.8992** | **0.8730** | **0.8229** | 0.7815 | 0.7972 |
| | LoRA | 0.8790 | 0.8999 | 0.8966 | 0.8694 | 0.8219 | **0.7824** | 0.7953 |
| | PiSSA | 0.8790 | 0.9008 | 0.8957 | 0.8644 | 0.8101 | 0.7703 | **0.8009** |
| | ▲Best Impr. | +10.22% | +10.51% | +7.49% | +11.59% | +10.44% | +4.28% | +5.61% |
| | ▲Avg Impr. | +9.55% | +10.42% | +7.29% | +11.18% | +9.98% | +3.85% | +5.30% |
| SQuAD | Base | 0.6833 | 0.6121 | 0.6472 | 0.7032 | 0.6547 | 0.6715 | **0.6704** |
| | DoRA | **0.7614** | **0.7407** | 0.7560 | **0.7350** | 0.7234 | 0.7322 | 0.6445 |
| | LoRA | 0.7440 | 0.7328 | **0.7638** | 0.7247 | **0.7315** | **0.7503** | 0.6611 |
| | PiSSA | 0.7596 | 0.6858 | 0.7282 | 0.7348 | 0.7275 | 0.7403 | 0.6330 |
| | ▲Best Impr. | +7.81% | +12.86% | +11.66% | +3.18% | +7.68% | +7.88% | -0.93% |
| | ▲Avg Impr. | +7.17% | +10.77% | +10.21% | +2.83% | +7.28% | +6.94% | -2.42% |

Table 4: AUPR scores for hallucination detection baselines in SQuAD across three backbones.

| Model | Method | Semantic Consistency | | | Confidence | | Entropy | |
| | | SE | SC | Deg | MSP | Perplexity | Mean Ent. | Pre Ent. |
|---|---|---|---|---|---|---|---|---|
| LLaMA-3.2-3B-Instruct | Base | 0.2456 | 0.2110 | 0.1601 | 0.1952 | 0.1849 | 0.1845 | **0.2125** |
| | DoRA | 0.2951 | **0.3433** | 0.1961 | 0.2219 | 0.2044 | 0.2092 | 0.1873 |
| | LoRA | **0.3400** | 0.3312 | **0.2773** | **0.3101** | **0.2652** | **0.2754** | 0.1751 |
| | PiSSA | 0.2640 | 0.2340 | 0.1853 | 0.2186 | 0.1951 | 0.2055 | 0.1242 |
| | ▲Best Impr. | +9.44% | +13.23% | +11.72% | +11.49% | +8.03% | +9.07% | -2.52% |
| | ▲Avg Impr. | +5.41% | +9.18% | +5.95% | +5.50% | +3.67% | +4.55% | -5.03% |
| Mistral-7B-Instruct-v0.3 | Base | 0.1875 | 0.1288 | 0.1608 | 0.1736 | 0.1386 | 0.1503 | **0.1478** |
| | DoRA | **0.2519** | **0.2758** | **0.2773** | **0.2618** | **0.1983** | **0.1992** | 0.1195 |
| | LoRA | 0.2176 | 0.2294 | 0.2628 | 0.1969 | 0.1747 | 0.1898 | 0.1253 |
| | PiSSA | 0.2516 | 0.2190 | 0.2403 | 0.2121 | 0.1784 | 0.1933 | 0.1071 |
| | ▲Best Impr. | +6.44% | +14.70% | +11.65% | +8.82% | +5.97% | +4.89% | -2.25% |
| | ▲Avg Impr. | +5.29% | +11.26% | +9.93% | +5.00% | +4.52% | +4.38% | -3.05% |
| Qwen2.5-3B-Instruct | Base | 0.1787 | 0.1767 | 0.2167 | 0.1658 | 0.1534 | 0.1578 | 0.1348 |
| | DoRA | 0.3021 | 0.2659 | 0.3172 | **0.2572** | 0.2528 | 0.2587 | 0.1408 |
| | LoRA | 0.2635 | 0.2665 | **0.3330** | 0.2198 | 0.2603 | 0.2762 | **0.1549** |
| | PiSSA | **0.3113** | **0.2952** | 0.3214 | 0.2282 | **0.2622** | **0.2816** | 0.1391 |
| | ▲Best Impr. | +13.26% | +11.85% | +11.63% | +9.14% | +10.88% | +12.38% | +2.01% |
| | ▲Avg Impr. | +11.36% | +9.92% | +10.72% | +6.93% | +10.50% | +11.44% | +1.01% |

and confidence methods aggregate uncertainty at the meaning level. They align more directly with whether

the model "knows" the answer. PEFT fine-tuning on QA tasks encourages the model to produce semantically coherent responses, thereby improving semantic-level calibration without reducing token-level entropy.

Additionally, **we report AUPR scores as a more informative metric for SQuAD in Table 4 due to severe class imbalance in SQuAD (high accuracy, low hallucination rate).** PEFT also improves hallucination detection performance systematically on SQuAD across all models observed from AUPR consistent with AUROC. These results confirm that PEFT enhances hallucination detectability even under imbalanced class conditions where AUROC may be misleadingly optimistic. To complement this SQuAD-specific analysis, we also report AUPR results for TriviaQA and NQ-Open in Appendix Tables 9 and 10, which help contextualize why AUROC gains in open-domain QA do not always translate into equally consistent AUPR gains.

💡 **Takeaway# 3: PEFT improves the performance of hallucination detectors by shifting scores away from the overconfident regime.** The distribution plots (Figure 3) of uncertainty scores place heavy probability mass concentrated near the lower bound of the score range (full results in Appendix E; see Figures 8 and 9). We further verify that such concentration reflects systematic overconfidence of hallucination detectors when applied to the base model. Specifically, we partition the model outputs into four quadrants based on uncertainty level and correctness (Figure 4). The right is the correct zone while the left is the incorrect zone. The top row is the unconfident zone while the bottom row is the confident zone. The uncertainty threshold for each model is set to its own median uncertainty score. We define the **reliability ratio** as safe / (safe + cautious) and the **danger ratio** as danger / (danger + detectable).

As shown in Figure 4 (Right), the danger ratio, which reflects overconfidence, is at a high level of 34.7%. After PEFT, it decreases consistently: reliability increases from 65.11% (Base) to 72.74% (PiSSA), indicating that responses become obviously more trustworthy after PEFT, while the danger ratio also decreases by 5% (PiSSA) at most. Appendix F extends this analysis across datasets, and Figure 11 shows the same qualitative shift on NQ-Open. Conversely, the reliability ratio increases after PEFT, indicating that correct outputs are more often assigned low uncertainty. Together, these shifts help explain the performance gains of hallucination detection methods under PEFT.

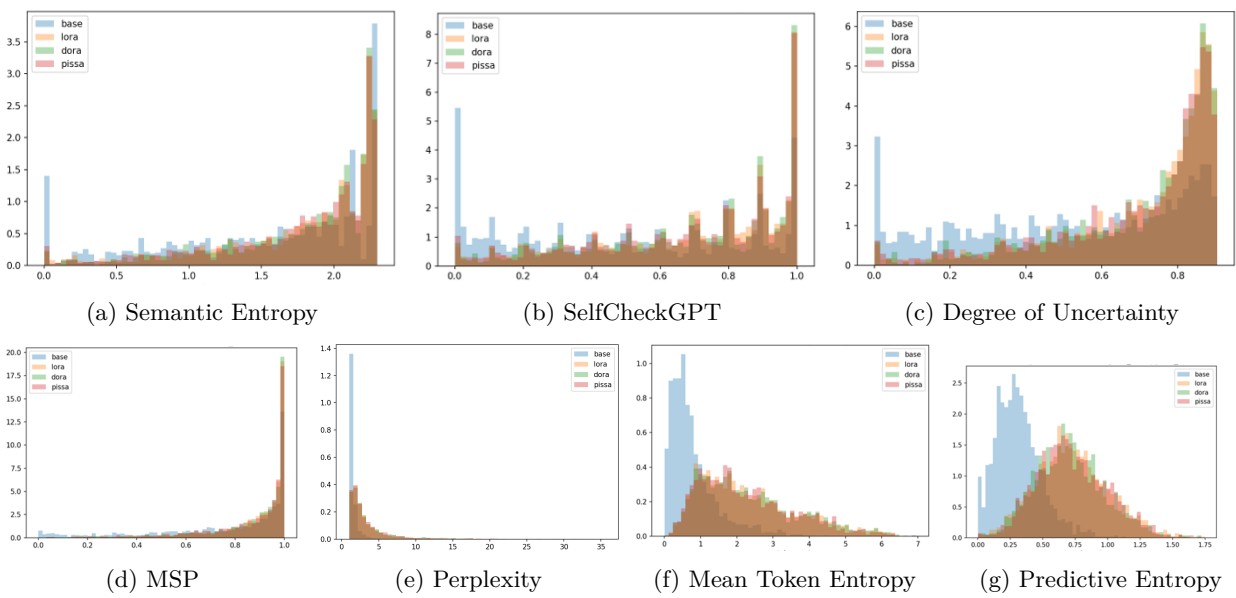

Figure 3: Uncertainty score density distributions across PEFT methods on Qwen-NQ-Open. X-axis represents the uncertainty score.

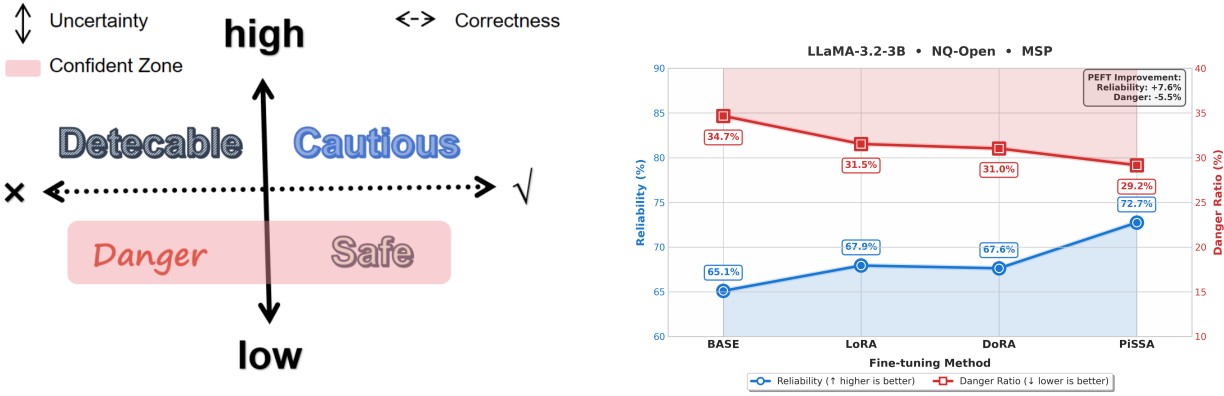

Figure 4: (Left) Uncertainty-correctness quadrant. (Right) Uncertainty-correctness analysis on NQ-Open using MSP (Llama-3.2-3B).

## 4 Behavior Analysis

### 4.1 Statistical Analysis: Track Dangerous to Detectable Migration

We track the dangerous cases of random three groups under the base model, and examine how they change after different PEFT fine-tuning methods across three datasets and two current prime and SOTA hallucination detectors (Semantic Entropy and SelfCheckGPT) on Llama-3.2-3B-Instruct. Table 5 presents the migration effects of PEFT methods. We define three metrics: **Corr. R.** (Correct / Total Danger) measures dangerous outputs corrected by PEFT; **Detect. R.** (Detectable / Total Danger) measures dangerous outputs becoming detectable; and **Conv. R.** (Detectable / (Detectable + Still Danger)) measures conversion efficiency from undetectable to detectable errors.

💡 **Takeaway#4: PiSSA as the best safety protector. DoRA as the most effective knowledge corrector in open-domain QA while LoRA achieves the consistent and obvious great performance to rectify the dangers on SQuAD.**

Firstly, we notice that PiSSA achieved the highest average detectable ratio. This indicates that PiSSA is particularly effective at calibrating the model's uncertainty. Secondly, in open-domain QA datasets, including TriviaQA and NQ-open, DoRA shows the best performance to directly fix the factual erros, achieving the highest correct rate (44.3% with SCG on TriviaQA ). It shows DoRA's weight-decomposed updates may be more efficient at injecting or activating correct knowledge, and better support knowledge retrieval from parametric memory. Conversely, LoRA dominates on SQuAD. LoRA achieves the highest correct rate (45.76%). This pattern is consistent with the observation in Figure 10. We find that on SQuAD, LoRA always reduces the danger ratio even by around 50% significantly. We guess that LoRA's parameter-efficient updates excel at improving attention based extraction.

**Error–success transitions.** While the preceding analysis tracks *ground-truth correctness* transitions among dangerous cases, we now examine *detection outcome* transitions to understand how PEFT reshapes detector performance at the instance level. Table 16 (Appendix) reports case-transition statistics on TriviaQA using LLaMA-3.2-3B-Instruct with Semantic Entropy. Across all three PEFT methods, the number of Error→Correct transitions substantially exceeds the number of Correct→Error transitions. For example, DoRA converts 506 previous detection errors into correct detections while introducing only 214 new errors, yielding a net gain consistent with the large AUROC improvement from 0.675 to 0.872.

### 4.2 Case study

We show two typical cases generated from LLaMA-3.2-3B in the boxes below to illustrate the two hallucination behaviors. Their unceratainty scores are calculated by SelfCheckGPT. Additionally, the first case is from

Table 5: Safety decomposition on **LLaMA-3.2-3B-Instruct** (%). Blue/pink = best correctness/detectability per block.

| Dataset | PEFT | SelfCheckGPT | | | Semantic Entropy | | |
|---|---|---|---|---|---|---|---|
| | | Correctness | Detectability | Conversion | Correctness | Detectability | Conversion |
| NQ-Open | LoRA | 20.2 | 30.3 | 38.0 | 26.3 | 32.6 | 44.1 |
| | PiSSA | 21.0 | **37.1** | **46.9** | 26.3 | **33.2** | **45.1** |
| | DoRA | **21.7** | 34.1 | 43.5 | **27.6** | 30.6 | 42.2 |
| TriviaQA | LoRA | 38.6 | 34.1 | 55.6 | 29.9 | 41.5 | 59.2 |
| | PiSSA | 30.7 | 34.1 | 55.7 | 29.9 | **45.7** | 65.2 |
| | DoRA | **44.3** | 34.1 | **61.2** | **33.3** | 44.4 | **66.7** |
| SQuAD | LoRA | **45.8** | 37.3 | 68.8 | **37.7** | 50.9 | 81.8 |
| | PiSSA | 42.4 | **47.5** | **82.4** | 35.9 | **52.8** | **82.4** |
| | DoRA | 37.3 | 45.8 | 73.0 | 35.9 | 45.3 | 70.6 |

NQ-Open and the second is from SQuAD. Case 1 demonstrates how PEFT handles a factual recall question while case 2 shows an extractive QA scenario.

### 4.3 Response Length as a Control

A simpler concern is whether PEFT improves detection merely because fine-tuned models produce shorter responses. Table 11 (Appendix H) confirms that PEFT-tuned models do generate substantially shorter answers: for example, on NQ-Open the mean length drops from 11.92 tokens (Base) to roughly 5.3 tokens across all three PEFT methods. However, the detection gains are concentrated in semantic-consistency-based and confidence-based detectors (e.g., SelfCheckGPT, Semantic Entropy, MSP), whereas token-level entropy detectors—which would be most directly affected by length reduction—show only marginal or inconsistent improvement. This dissociation indicates that the detectability gains reported above cannot be attributed solely to shorter outputs, and instead reflect genuine improvements in uncertainty calibration. The same qualitative pattern is robust to sampling temperature and PEFT configuration choices; Appendices I and J show that the main detectability gains persist across temperatures, ranks, and target modules.

## 5 White-Box Detector: An Uncertainty Probing

While the unsupervised black-box detectors (semantic consistency based and confidence based) benefit from PEFT, it remains unclear whether this improvement extends to other detection paradigms. We examine

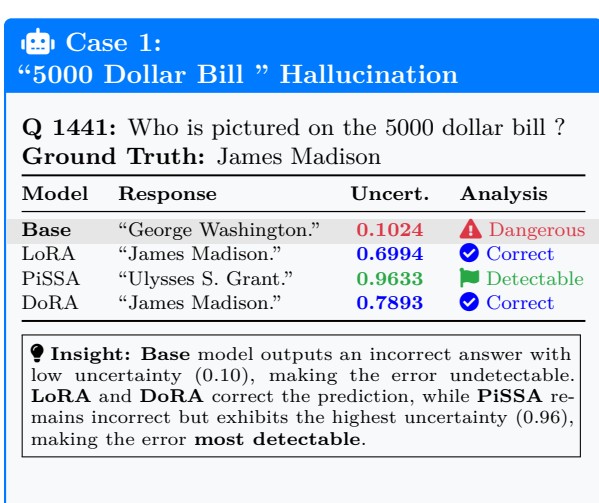

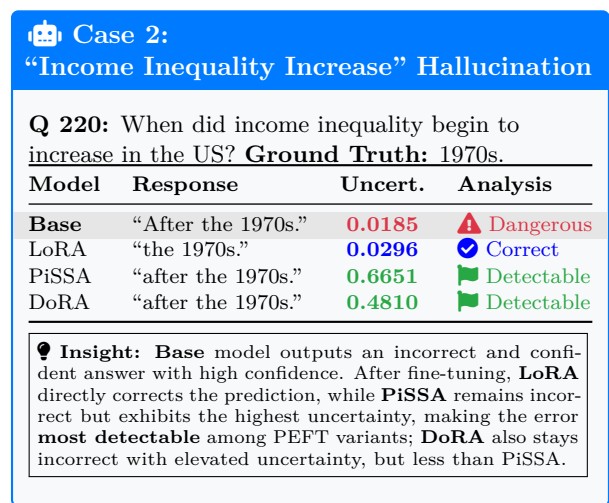

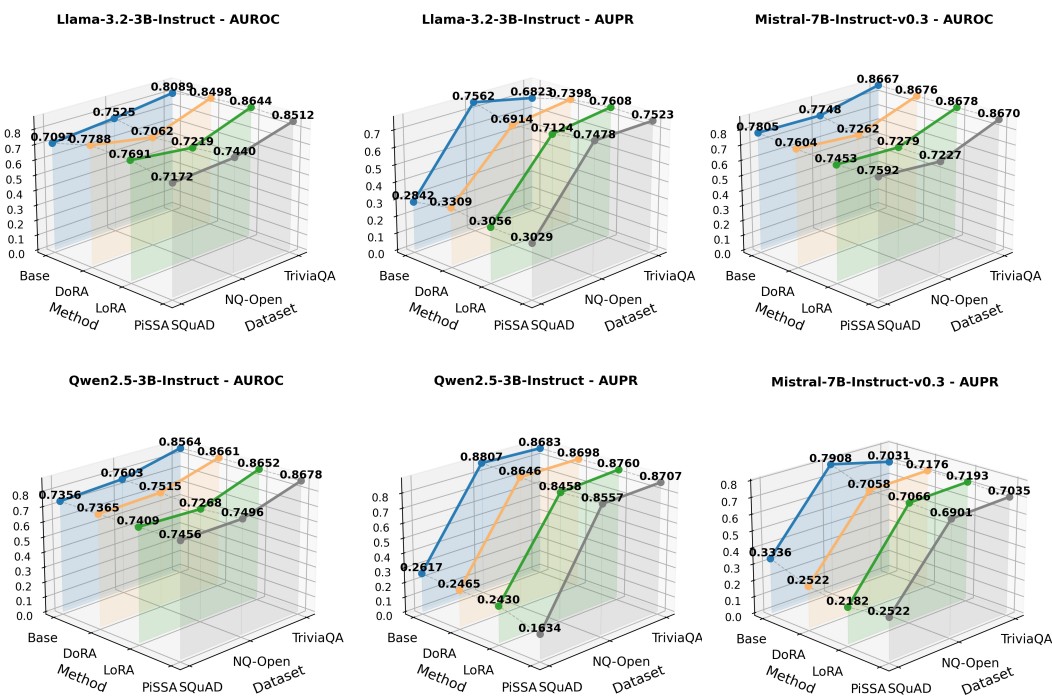

Figure 5: **Probe hallucination detection scores.** We train logistic regression probes on every layer's hidden states to detect hallucination, reporting the best-performing layer's AUROC and AUPR on the validation set. Z axis shows the AUROC/AUPR.

linear probing, a supervised white-box approach that directly classifies hidden representations to test whether PEFT universally enhances hallucination detectability.

💡**Takeaway#5: PEFT raises black-box hallucination detectors but disrupts linear probe based detector.**

Figure 5 and Table 6 reveal striking inconsistency: on TriviaQA, PEFT improves probing performance; on NQ-Open, PEFT uniformly degrades it; on SQuAD, effects vary by model. This inconsistency contrasts sharply with black-box unsupervised uncertainty detectors (Table 1, 2 and Table 4), which improve consistently almost across all settings after PEFT. The PCA visualizations in Appendix Figure 12 confirm this interpretation. After PEFT, hidden representations do not exhibit clearer separation between correct and incorrect predictions—in some cases, clusters become less distinguishable. Yet downstream detection via black-box methods improves substantially. By table 6, these results support Takeaway #5 by showing that, unlike black-box uncertainty detectors, probe-based methods do not consistently benefit from PEFT and may even degrade, highlighting a representational shift induced by fine-tuning.

The divergence suggests a critical insight that supervised probing based hallucination detectors proves unreliable for PEFT-tuned models compared with the unsupervised hallucination detectors. We conjecture that PEFT redistributes uncertainty from linearly separable hidden representations to output level behaviors, disrupting the geometric structure that probes exploit. Characterizing this representational shift and developing probe methods robust to fine-tuning remains a promising direction for future work.

## 6 Conclusions, Limitations, and Future Directions

To our knowledge, this work presents the first systematic study of how PEFT affects answer-level hallucination detection in LLMs for fact-seeking QA. Across three backbones, three PEFT methods, three QA benchmarks, and multiple answer-level detectors, we find that PEFT yields only modest gains in answer accuracy but consistently improves the uncertainty-aware of externally verified incorrect answers, especially for semantic-

consistency-based and confidence-based detectors, whereas token-level entropy-based detectors improve less consistently. These results suggest that, in this setting, PEFT primarily reshapes detector-relevant uncertainty signals rather than substantially expanding factual knowledge, while its effects on supervised white-box probing remain mixed. Appendix K further shows that full supervised fine-tuning exhibits a similar qualitative detectability-improvement pattern, suggesting that this effect is not unique to PEFT, although PEFT remains practically attractive because it approaches this behavior while updating only a small fraction of parameters. Our analysis is specific to answer-level detection in fact-seeking QA, and its generalization to other tasks, model scales, and adaptation settings remains an open question. More broadly, future work should extend this framework to settings such as claim-level verification, long-form generation, reasoning-intensive tasks, code generation, multimodal problems, and interactive dialogue.

Table 6: Best-layer linear probe results (AUROC/AUPR) for hallucination detection.

| Model | Dataset | Method | Best Layer | AUROC | AUPR |
|---|---|---|---|---|---|
| LLaMA-3.2-3B-Instruct | NQ-Open | Base | 13 | **0.7525** | **0.7562** |
| | | DoRA | 13 | 0.7062 | 0.6914 |
| | | LoRA | 14 | 0.7219 | 0.7124 |
| | | PiSSA | 14 | 0.7440 | 0.7478 |
| | SQuAD | Base | 23 | 0.7097 | 0.2842 |
| | | DoRA | 14 | **0.7788** | **0.3309** |
| | | LoRA | 22 | 0.7691 | 0.3056 |
| | | PiSSA | 22 | 0.7172 | 0.3029 |
| | TriviaQA | Base | 18 | 0.8089 | 0.6823 |
| | | DoRA | 12 | 0.8498 | 0.7398 |
| | | LoRA | 16 | **0.8644** | **0.7608** |
| | | PiSSA | 12 | 0.8512 | 0.7523 |
| Mistral-7B-Instruct-v0.3 | NQ-Open | Base | 15 | **0.7748** | **0.7908** |
| | | DoRA | 14 | 0.7262 | 0.7058 |
| | | LoRA | 15 | 0.7279 | 0.7066 |
| | | PiSSA | 15 | 0.7227 | 0.6901 |
| | SQuAD | Base | 13 | **0.7805** | **0.3336** |
| | | DoRA | 22 | 0.7604 | 0.2522 |
| | | LoRA | 19 | 0.7453 | 0.2182 |
| | | PiSSA | 18 | 0.7592 | 0.2522 |
| | TriviaQA | Base | 14 | 0.8667 | 0.7031 |
| | | DoRA | 16 | 0.8676 | 0.7176 |
| | | LoRA | 16 | **0.8678** | **0.7193** |
| | | PiSSA | 16 | 0.8670 | 0.7035 |
| Qwen2.5-3B-Instruct | NQ-Open | Base | 24 | **0.7603** | **0.8807** |
| | | DoRA | 24 | 0.7515 | 0.8646 |
| | | LoRA | 24 | 0.7268 | 0.8458 |
| | | PiSSA | 24 | 0.7496 | 0.8557 |
| | SQuAD | Base | 26 | 0.7356 | **0.2617** |
| | | DoRA | 24 | 0.7365 | 0.2465 |
| | | LoRA | 24 | 0.7409 | 0.2430 |
| | | PiSSA | 34 | **0.7456** | 0.2434 |
| | TriviaQA | Base | 27 | 0.8564 | 0.8683 |
| | | DoRA | 24 | 0.8661 | 0.8698 |
| | | LoRA | 26 | 0.8652 | **0.8760** |
| | | PiSSA | 27 | **0.8678** | 0.8707 |

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

## A  Experimental Details

- For all the implemented LLMs, we prioritize using their officially provided application programming interface (API) if available. Besides, we ignore a little bit of questions that the LLMs refuse to answer. Otherwise, we deploy the model with LLMs on $8 \times$ A6000 GPUs.

- For instruction tuning, we use a consistent prompt: `"Answer the question in a short phrase.\n\nQuestion: {question}"` for closed-book QA, and prepend the passage for SQuAD.

- For TriviaQA and NQ-Open, we adopt a closed-book setting and input only the question, testing retrieval from parametric memory. For SQuAD, we provide both context and question, testing grounding in provided text. Since the official test sets for TriviaQA and NQ-Open often lack public groundtruth, we adopt a split-validation strategy on the validation sets. We use 2,500 samples for training, 500 for validation, and the remaining ∼2,000 (1,805 for NQ-Open) for testing.

- While the original evaluation protocol for TriviaQA is exact match score, we find it infeasible for generative language models, as the same answer may be represented with diversified texts. For one thing,For example, the question "What is the capital of the USA?" has the ground truth "Washington, D.C.". However, if the model answers "Washington" or "Washington DC", exact match may fail to recognize these as correct. For another, there are few questions that models refuse to answer which will results to a few additional effort to deal with the specific pattern to match that if using the EM/F1. In view of this, we use LLM-as-a-Judge for QA datasets. We deploy the most recently released LLM, GPT-5-mini, as our judging model and report the accuracy in percentile.

## B  LLM-as-Judge Consistency Analysis

In the main paper, we use GPT-5-mini as the judge. Following the reviewer's suggestion, we conduct a cross-judge consistency study. Specifically, we introduce two additional frontier models as independent judges alongside GPT-5-mini. We evaluate the first 200 test samples from TriviaQA, assessing answers generated by LLaMA-3.2-3B-Instruct.

**Judges.**  Azure GPT-5-mini / Claude 4.6 Opus / Gemini 3 Pro Preview

Table 7: Judge verdict distribution on TriviaQA (test split, first 200 samples).

| Judge | Correct | Incorrect |
|---|---|---|
| Azure GPT-5-mini | 67.50% | 32.50% |
| Claude 4.6 Opus | 69.67% | 30.33% |
| Gemini 3 Pro Preview | 66.87% | 33.13% |

Table 8: Pairwise agreement rates among judges.

| Judge Pair | Agreement Rate |
|---|---|
| Azure GPT-5-mini vs. Claude 4.6 Opus | 98.36% |
| Azure GPT-5-mini vs. Gemini 3 Pro Preview | 98.80% |
| Claude 4.6 Opus vs. Gemini 3 Pro Preview | 97.86% |
| All three judges agree | 97.73% |

Pairwise agreement exceeds 97% across all comparisons, suggesting that for factual QA tasks such as TriviaQA, frontier models apply highly consistent evaluation criteria.

## C  Fine-tuning Loss Curves

(Figure 6 and 7). This section reports training and evaluation loss curves for all PEFT configurations across models and datasets. The primary purpose of these figures is to verify the correctness and stability

of the fine-tuning process rather than to compare optimization efficiency across methods. As shown in Figures 6 and 7, LoRA, DoRA, and PiSSA consistently exhibit rapid convergence, smooth loss decay, and no signs of divergence or overfitting on LLaMA, Mistral, and Qwen backbones. Training and evaluation losses closely track each other across datasets, indicating that the observed differences in hallucination detection performance in the main paper are not artifacts of unstable training, optimization failure, or misconfiguration, but reflect genuine epistemic effects induced by PEFT.

## D  TriviaQA and NQ-Open hallucination detection AUPR score

(Table. 9 10). This section reports AUPR scores for TriviaQA and NQ-Open, complementing the AUROC analysis in the main paper. Because open-domain QA exhibits less severe class imbalance than SQuAD, these results illustrate why PEFT-induced improvements in uncertainty detectors do not always translate into consistent AUPR gains, supporting the discussion in Takeaway#2.

Table 9: AUPR scores for hallucination detection baselines on TriviaQA.

| Model | Method | Semantic Consistency | | | Confidence | | Entropy | |
|---|---|---|---|---|---|---|---|---|
| | | SE | SC | Deg | MSP | Perplexity | MeanEnt | PreEnt |
| LLaMA-3.2 -3B-Instruct | Base | 0.6750 | 0.7942 | 0.7669 | 0.6761 | 0.7245 | 0.7450 | 0.6198 |
| | DoRA | 0.7618 | 0.7768 | 0.7535 | 0.8011 | 0.7431 | 0.6907 | 0.6585 |
| | LoRA | 0.7155 | 0.7461 | 0.7030 | 0.7652 | 0.7071 | 0.6643 | 0.6198 |
| | PiSSA | **0.7782** | **0.8081** | **0.7686** | **0.8260** | **0.7826** | **0.7487** | **0.6865** |
| Mistral-7B -Instruct-v0.3 | Base | 0.5960 | 0.6181 | 0.6634 | 0.5536 | 0.4625 | **0.5187** | 0.5199 |
| | DoRA | **0.7289** | **0.7503** | **0.7263** | **0.7142** | 0.5899 | 0.5137 | 0.5216 |
| | LoRA | 0.6960 | 0.7462 | 0.7190 | 0.7068 | **0.5928** | 0.5185 | 0.5261 |
| | PiSSA | 0.6675 | 0.7241 | 0.6938 | 0.6922 | 0.5599 | 0.4905 | **0.5339** |
| Qwen2.5 -3B-Instruct | Base | 0.8437 | **0.8668** | **0.8758** | 0.8156 | 0.8265 | **0.8488** | **0.8049** |
| | DoRA | 0.8544 | 0.8660 | 0.8514 | 0.8986 | **0.8530** | 0.8088 | 0.7904 |
| | LoRA | **0.8591** | 0.8657 | 0.8472 | **0.9016** | 0.8525 | 0.8102 | 0.7774 |
| | PiSSA | 0.8433 | 0.8633 | 0.8519 | 0.9004 | 0.8377 | 0.7992 | 0.7771 |

Table 10: AUPR scores for hallucination detection baselines on NQ-Open.

| Model | Method | Semantic Consistency | | | Confidence | | Entropy | |
|---|---|---|---|---|---|---|---|---|
| | | SE | SC | Deg | MSP | Perplexity | MeanEnt | PreEnt |
| LLaMA-3.2 -3B-Instruct | Base | 0.6698 | 0.7459 | 0.7391 | 0.6758 | 0.7188 | 0.7203 | 0.6319 |
| | DoRA | 0.7062 | 0.7418 | 0.7070 | 0.7404 | 0.7097 | 0.6810 | 0.6558 |
| | LoRA | 0.7115 | 0.7412 | 0.7060 | **0.7470** | 0.7280 | 0.7003 | 0.6738 |
| | PiSSA | **0.7581** | **0.7908** | **0.7619** | 0.7803 | **0.7628** | **0.7372** | **0.7146** |
| Mistral-7B -Instruct-v0.3 | Base | 0.6920 | 0.7600 | **0.7759** | 0.6742 | 0.6400 | 0.6617 | 0.6393 |
| | DoRA | 0.7193 | 0.7682 | 0.7515 | 0.7435 | 0.6689 | 0.6406 | 0.6810 |
| | LoRA | **0.7284** | **0.7716** | 0.7532 | **0.7495** | **0.6809** | **0.6522** | **0.6930** |
| | PiSSA | 0.7281 | 0.7708 | 0.7483 | 0.7394 | 0.6660 | 0.6444 | 0.6743 |
| Qwen2.5 -3B-Instruct | Base | **0.8769** | 0.8904 | **0.8964** | 0.8428 | **0.8532** | **0.8603** | 0.8265 |
| | DoRA | 0.8620 | **0.9049** | 0.8811 | 0.8770 | 0.8407 | 0.8154 | **0.8385** |
| | LoRA | 0.8557 | 0.9003 | 0.8771 | **0.8779** | 0.8418 | 0.8161 | 0.8391 |
| | PiSSA | 0.8538 | 0.8902 | 0.8789 | 0.8859 | 0.8385 | 0.8095 | 0.8309 |

# E   The Uncertainty Score Density

(Figure 8 and Figure 9). This section visualizes the distribution of uncertainty scores produced by different hallucination detectors before and after PEFT. These density plots provide qualitative evidence for Takeaway #3, showing how PEFT disperses degenerate near-zero uncertainty mass and restores meaningful separation between correct and incorrect responses.

# F   The reliability and danger ratio on Llama-3.2-3B across all datasets

(Table. 10 11). This section provides a comprehensive confidence and correctness analysis across datasets using reliability and danger ratios. The figures quantify how PEFT shifts errors from confident hallucinations to detectable uncertain cases, offering behavioral evidence that PEFT improves safety-relevant properties beyond aggregate AUROC metrics.

# G   PCA plots of hidden state in layer

(Figure. 12). This section presents PCA visualizations of hidden representations at the best probing layers. The plots provide geometric intuition for the probe results, illustrating that PEFT does not necessarily increase linear separability between correct and incorrect answers, even when black-box hallucination detection improves.

# H   Response Length Analysis

This section provides the full generation length statistics referenced in the main text. We measure the token counts of model outputs on the test split of three datasets using LLaMA-3.2-3B-Instruct.

Table 11: Generation length statistics across datasets and PEFT methods (LLaMA-3.2-3B-Instruct, test split).

| Dataset | Method | Mean | Med. | Std | Min | Max | Q25 | Q75 |
|---------|--------|------|------|-----|-----|-----|-----|-----|
| TriviaQA | Base | 7.07 | 5.0 | 4.30 | 2 | 20 | 4.0 | 8.0 |
| TriviaQA | LoRA | 4.66 | 4.0 | 1.61 | 3 | 18 | 4.0 | 5.0 |
| TriviaQA | DoRA | 4.68 | 4.0 | 1.58 | 3 | 17 | 4.0 | 5.0 |
| TriviaQA | PiSSA | 4.98 | 5.0 | 1.54 | 3 | 19 | 4.0 | 6.0 |
| NQ-Open | Base | 11.92 | 13.0 | 5.02 | 3 | 20 | 7.0 | 16.0 |
| NQ-Open | LoRA | 5.30 | 5.0 | 1.78 | 3 | 16 | 4.0 | 6.0 |
| NQ-Open | DoRA | 5.29 | 5.0 | 1.77 | 3 | 16 | 4.0 | 6.0 |
| NQ-Open | PiSSA | 5.39 | 5.0 | 1.83 | 3 | 17 | 4.0 | 6.0 |
| SQuAD | Base | 9.29 | 8.0 | 4.72 | 2 | 22 | 5.0 | 12.0 |
| SQuAD | LoRA | 6.01 | 5.0 | 2.93 | 3 | 22 | 4.0 | 7.0 |
| SQuAD | DoRA | 6.39 | 5.0 | 3.17 | 3 | 22 | 4.0 | 7.0 |
| SQuAD | PiSSA | 6.41 | 5.0 | 3.26 | 3 | 22 | 4.0 | 7.0 |

Across all three datasets, PEFT-tuned models generally generate shorter responses than the base model. This pattern is especially pronounced on TriviaQA and NQ-Open, where the mean and median generation lengths decrease substantially after PEFT. The reduced response length may partially reflect more concise answer generation behavior after task adaptation.

# I Temperature Sensitivity

We conduct a robustness check across different sampling temperatures. Specifically, we report AUROC results for the base model and all three PEFT methods at temperatures 0.6, 0.8, and 1.0. We include only detectors that require sampling multiple responses, since deterministic detectors are unaffected by temperature.

Overall, PEFT-induced improvements persist across all tested temperatures, supporting the robustness of our main conclusions.

Table 12: Hallucination detection performance across PEFT methods and sampling temperatures. Results are reported in AUROC.

| Configuration | Degree of Uncertainty | Predictive Entropy | SelfCheckGPT | Semantic Entropy |
|---|---|---|---|---|
| Base, $T$=0.6 | 0.8500 | 0.7793 | 0.8419 | 0.8155 |
| LoRA, $T$=0.6 | 0.8603 | 0.7943 | 0.8596 | 0.8524 |
| DoRA, $T$=0.6 | 0.8921 | 0.8072 | 0.8886 | 0.8717 |
| PiSSA, $T$=0.6 | 0.8821 | 0.8086 | 0.8813 | 0.8759 |
| Base, $T$=0.8 | 0.8661 | 0.7815 | 0.8620 | 0.8259 |
| LoRA, $T$=0.8 | 0.8831 | 0.7891 | 0.8643 | 0.8514 |
| DoRA, $T$=0.8 | 0.8914 | 0.8003 | 0.8955 | 0.8822 |
| PiSSA, $T$=0.8 | 0.8841 | 0.8112 | 0.8880 | 0.8686 |
| Base, $T$=1.0 | 0.8703 | 0.7819 | 0.8737 | 0.8205 |
| LoRA, $T$=1.0 | 0.8779 | 0.8216 | 0.8889 | 0.8737 |
| DoRA, $T$=1.0 | 0.8738 | 0.8138 | 0.8838 | 0.8823 |
| PiSSA, $T$=1.0 | 0.8772 | 0.8087 | 0.8896 | 0.8797 |

# J Rank Sensitivity and MLP Inclusion Tests

To verify that the detectability improvement is not a narrow artifact of a single PEFT configuration, we evaluate ranks 8, 16, and 32 for three PEFT methods (LoRA, DoRA, and PiSSA), targeting both attention and MLP projection layers on TriviaQA using LLaMA-3.2-3B-Instruct. Results are reported in Tables 13 and 14, and all metrics are AUROC. Overall, detection improvements are consistent across ranks 8, 16, and 32. These results further support our main takeaways that PEFT improves semantic-consistency-based and confidence-based hallucination detectors. Table 13 shows that this pattern is stable across different ranks, while Table 14 shows that it also holds when comparing *Attention*-only tuning against *MLP+Attn.*

These results show that the gains are not tied to a single rank choice. In addition, tuning attention modules alone already yields strong improvements, while extending tuning to both MLP and attention layers preserves the same qualitative conclusions.

Table 13: Hallucination detection performance across PEFT configurations (rank sensitivity). All results are AUROC on TriviaQA with LLaMA-3.2-3B-Instruct. In the first column, each configuration specifies the PEFT method, rank, and target modules.

| Configuration | SE | SC | Deg | MSP | Perplexity | MeanEnt | PreEnt |
|---|---|---|---|---|---|---|---|
| Base | 0.8205 | 0.8737 | 0.8703 | 0.8154 | 0.8328 | 0.8457 | 0.7819 |
| LoRA 8, MLP+Attn | 0.8711 | 0.8807 | 0.8726 | 0.8742 | 0.8574 | 0.8334 | 0.8063 |
| LoRA 16, MLP+Attn | 0.8743 | 0.8801 | 0.8737 | 0.8707 | 0.8546 | 0.8344 | 0.8037 |
| LoRA 32, MLP+Attn | 0.8703 | 0.8798 | 0.8728 | 0.8698 | 0.8522 | 0.8256 | 0.7980 |
| DoRA 8, MLP+Attn | 0.8778 | 0.8927 | 0.8836 | 0.8827 | 0.8615 | 0.8322 | 0.8094 |
| DoRA 16, MLP+Attn | 0.8748 | 0.8857 | 0.8796 | 0.8734 | 0.8492 | 0.8236 | 0.7968 |
| DoRA 32, MLP+Attn | 0.8759 | 0.8881 | 0.8745 | 0.8757 | 0.8529 | 0.8252 | 0.8042 |
| PiSSA 8, MLP+Attn | 0.8726 | 0.8897 | 0.8814 | 0.8832 | 0.8602 | 0.8273 | 0.8115 |
| PiSSA 16, MLP+Attn | 0.8721 | 0.8784 | 0.8712 | 0.8763 | 0.8578 | 0.8358 | 0.8167 |
| PiSSA 32, MLP+Attn | 0.8698 | 0.8794 | 0.8797 | 0.8694 | 0.8570 | 0.8340 | 0.8098 |

Table 14: Hallucination detection performance across PEFT methods and target modules. All results are AUROC on TriviaQA with LLaMA-3.2-3B-Instruct.

| Configuration | Modules | SE | SC | Deg | MSP | Perplexity | MeanEnt | PreEnt |
|---|---|---|---|---|---|---|---|---|
| Base | — | 0.8205 | 0.8737 | 0.8703 | 0.8154 | 0.8328 | 0.8457 | 0.7819 |
| LoRA | Attn only | 0.8737 | 0.8889 | 0.8779 | 0.8656 | 0.8597 | 0.8303 | 0.8216 |
| LoRA | MLP+Attn | 0.8613 | 0.8798 | 0.8728 | 0.8648 | 0.8462 | 0.8256 | 0.7880 |
| DoRA | Attn only | 0.8823 | 0.8838 | 0.8738 | 0.8866 | 0.8605 | 0.8202 | 0.8138 |
| DoRA | MLP+Attn | 0.8759 | 0.8881 | 0.8745 | 0.8757 | 0.8529 | 0.8252 | 0.8042 |
| PiSSA | Attn only | 0.8797 | 0.8896 | 0.8772 | 0.8878 | 0.8595 | 0.8275 | 0.8087 |
| PiSSA | MLP+Attn | 0.8698 | 0.8794 | 0.8697 | 0.8694 | 0.8570 | 0.8340 | 0.8098 |

## K   Full Supervised Fine-Tuning (SFT) Baseline

To contextualize the PEFT results presented in the main text, we report hallucination detection performance after standard full supervised fine-tuning (SFT) on the same QA training sets. Table 15 presents matched base and SFT AUROC/AUPR results for seven black-box detectors across three models and three datasets. Comparing these results with the PEFT tables in the main text allows readers to assess whether the detectability improvements observed under PEFT are unique to parameter-efficient methods or also emerge under full fine-tuning.

Table 15: Base and SFT hallucination detection results. Each cell reports AUROC/AUPR. Within each dataset block, each backbone is shown with its base row followed by its full-SFT row.

| Configuration | SE | SC | Deg | MSP | Perplexity | MeanEnt | PreEnt |
|---|---|---|---|---|---|---|---|
| **TriviaQA** | | | | | | | |
| LLaMA-3B Base | 0.821/0.675 | 0.874/0.794 | 0.870/0.767 | 0.815/0.676 | 0.833/0.725 | 0.846/0.745 | 0.782/0.620 |
| LLaMA-3B SFT | 0.887/0.791 | 0.883/0.797 | 0.873/0.767 | 0.882/0.813 | 0.837/0.715 | 0.811/0.688 | 0.802/0.676 |
| Qwen-3B Base | 0.832/0.844 | 0.855/0.867 | 0.855/0.876 | 0.813/0.816 | 0.816/0.827 | 0.832/0.849 | 0.781/0.805 |
| Qwen-3B SFT | 0.890/0.870 | 0.892/0.868 | 0.877/0.842 | 0.902/0.896 | 0.856/0.838 | 0.823/0.797 | 0.791/0.766 |
| Mistral-7B Base | 0.787/0.596 | 0.797/0.618 | 0.824/0.663 | 0.757/0.554 | 0.719/0.463 | 0.740/0.519 | 0.745/0.520 |
| Mistral-7B SFT | 0.841/0.843 | 0.846/0.848 | 0.846/0.844 | 0.832/0.840 | 0.774/0.766 | 0.743/0.734 | 0.781/0.784 |
| **NQ-Open** | | | | | | | |
| LLaMA-3B Base | 0.703/0.670 | 0.743/0.746 | 0.738/0.739 | 0.698/0.676 | 0.726/0.719 | 0.730/0.720 | 0.665/0.632 |
| LLaMA-3B SFT | 0.774/0.776 | 0.797/0.805 | 0.778/0.779 | 0.797/0.798 | 0.751/0.762 | 0.729/0.739 | 0.751/0.768 |
| Qwen-3B Base | 0.767/0.877 | 0.776/0.890 | 0.791/0.896 | 0.719/0.843 | 0.717/0.853 | 0.723/0.860 | 0.692/0.827 |
| Qwen-3B SFT | 0.820/0.874 | 0.837/0.885 | 0.822/0.876 | 0.831/0.880 | 0.784/0.839 | 0.742/0.811 | 0.766/0.828 |
| Mistral-7B Base | 0.705/0.692 | 0.736/0.760 | 0.758/0.776 | 0.679/0.674 | 0.650/0.640 | 0.668/0.662 | 0.666/0.639 |
| Mistral-7B SFT | 0.787/0.861 | 0.794/0.878 | 0.765/0.853 | 0.807/0.887 | 0.760/0.844 | 0.726/0.818 | 0.753/0.844 |
| **SQuAD** | | | | | | | |
| LLaMA-3B Base | 0.716/0.246 | 0.698/0.211 | 0.676/0.160 | 0.697/0.195 | 0.643/0.185 | 0.638/0.184 | 0.681/0.212 |
| LLaMA-3B SFT | 0.780/0.249 | 0.747/0.295 | 0.767/0.279 | 0.742/0.206 | 0.731/0.217 | 0.744/0.235 | 0.670/0.141 |
| Qwen-3B Base | 0.667/0.179 | 0.611/0.177 | 0.660/0.217 | 0.668/0.166 | 0.652/0.153 | 0.656/0.158 | 0.631/0.135 |
| Qwen-3B SFT | 0.809/0.305 | 0.774/0.286 | 0.806/0.311 | 0.741/0.319 | 0.738/0.320 | 0.742/0.324 | 0.663/0.153 |
| Mistral-7B Base | 0.683/0.188 | 0.612/0.129 | 0.647/0.161 | 0.703/0.174 | 0.655/0.139 | 0.671/0.150 | 0.670/0.148 |
| Mistral-7B SFT | 0.764/0.328 | 0.727/0.351 | 0.758/0.381 | 0.771/0.265 | 0.754/0.271 | 0.763/0.295 | 0.677/0.234 |

**Comparison with Full SFT.**   Comparing Table 15 with the PEFT results in the main text, a similar qualitative detectability-improvement pattern also appears under full SFT, suggesting that the effect is not unique to parameter-efficient tuning. In many detector–dataset pairs, SFT yields comparable or slightly higher scores, which is consistent with a broader effect of task-specific supervision on uncertainty signals. At the same time, PEFT remains practically attractive because it approaches this pattern while updating only a small fraction of parameters. Overall, these results suggest that the uncertainty reshaping documented in our main findings is driven primarily by exposure to task-specific supervision rather than by the particular parameterization of the adaptation method.

## L   Error–Success Transitions

We further examine instance-level detection transitions on TriviaQA using LLaMA-3.2-3B-Instruct and Semantic Entropy. Table 16 reports how many base-model detection errors become correct after PEFT, together with how many previously correct detections become errors.

Table 16: Error/success transitions on TriviaQA using LLaMA-3.2-3B-Instruct and Semantic Entropy. Base AUROC = 0.675; base detection errors = 666 (34.7%).

| PEFT | AUROC | Errors→Correct | Correct→Errors |
|------|-------|----------------|----------------|
| LoRA | 0.857 | 518 | 213 |
| PiSSA | 0.860 | 506 | 269 |
| DoRA | **0.872** | 506 | 214 |

All three PEFT methods convert substantially more detection errors into correct decisions than the reverse, confirming that the AUROC gains are driven by broad instance-level improvements rather than a few isolated score changes. LoRA yields the largest number of error-to-correct transitions, while DoRA achieves the best overall AUROC with nearly the same number of recovered errors but far fewer regressions than PiSSA. This pattern suggests that DoRA provides the best balance between fixing base-model detection failures and avoiding newly introduced errors.

## M  Ethical Considerations

We used AI assistants (such as Claude, ChatGPT and Gemini) during the research process for: (1) polishing and editing the manuscript for clarity and grammar; (2) providing suggestions for figure design and visualization; and (3) assisting with code implementation and debugging. All AI-generated content was carefully reviewed, verified, and revised by the authors, who take full responsibility for the final manuscript.

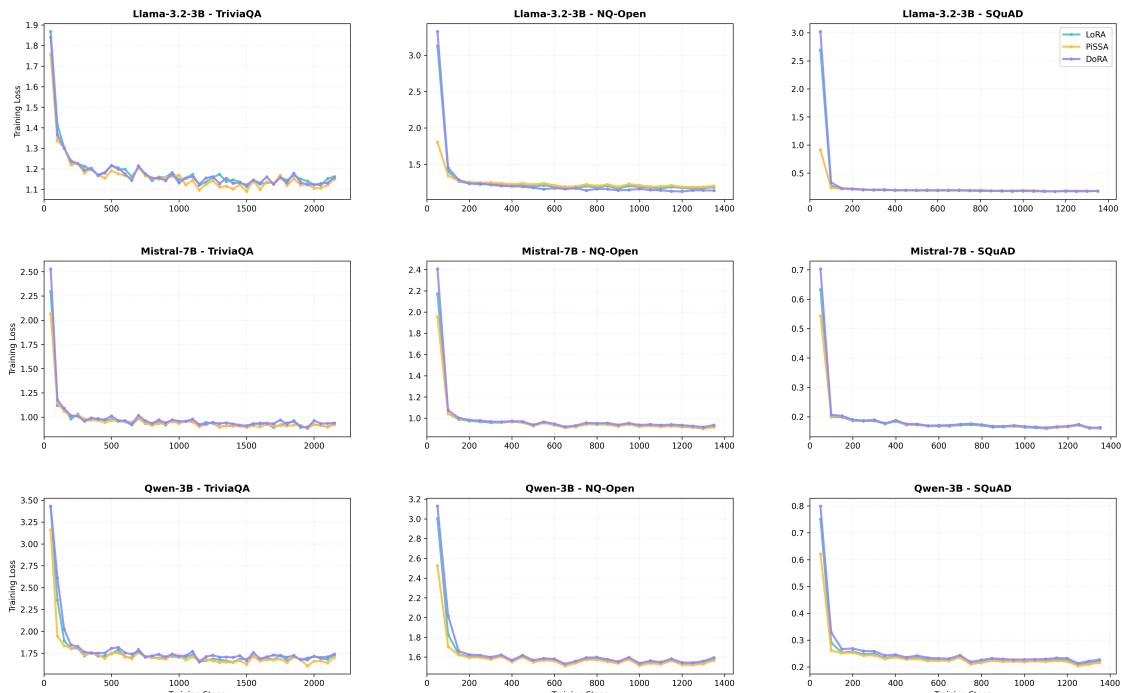

Figure 6: **Training loss curves.**

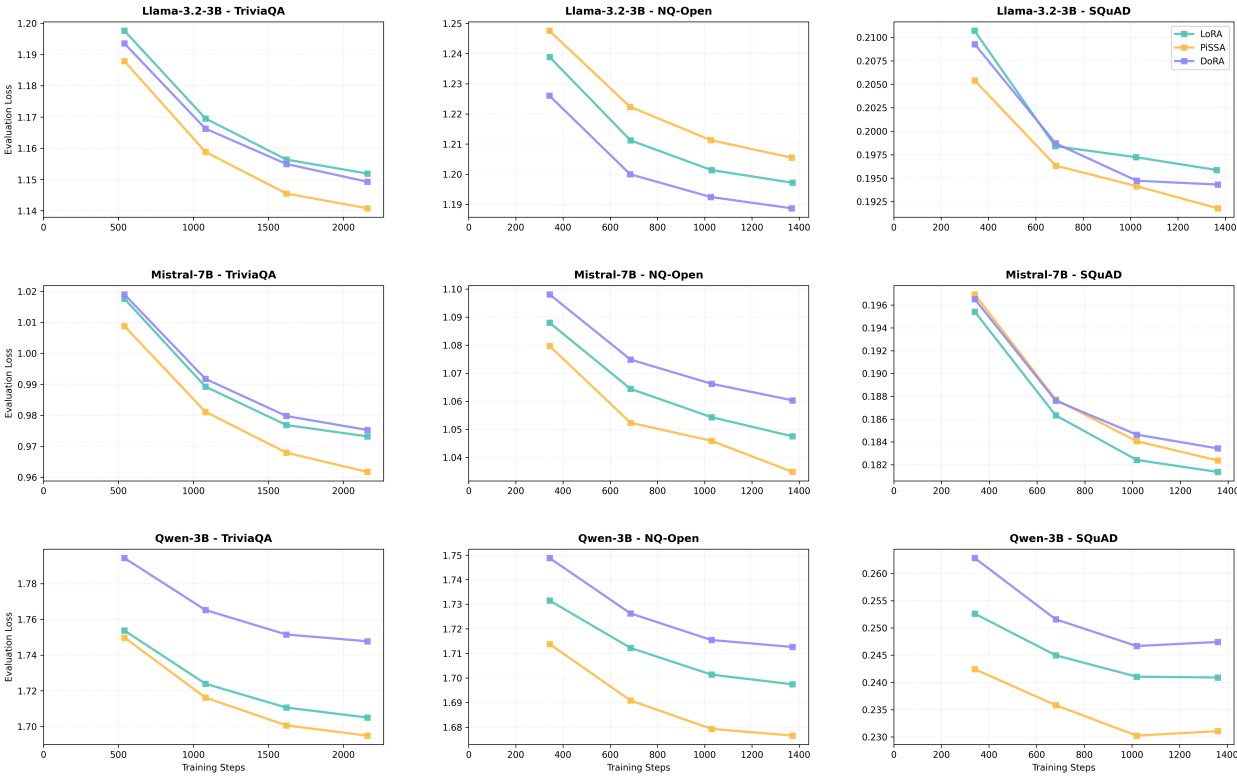

Figure 7: **Evaluation loss curves.**

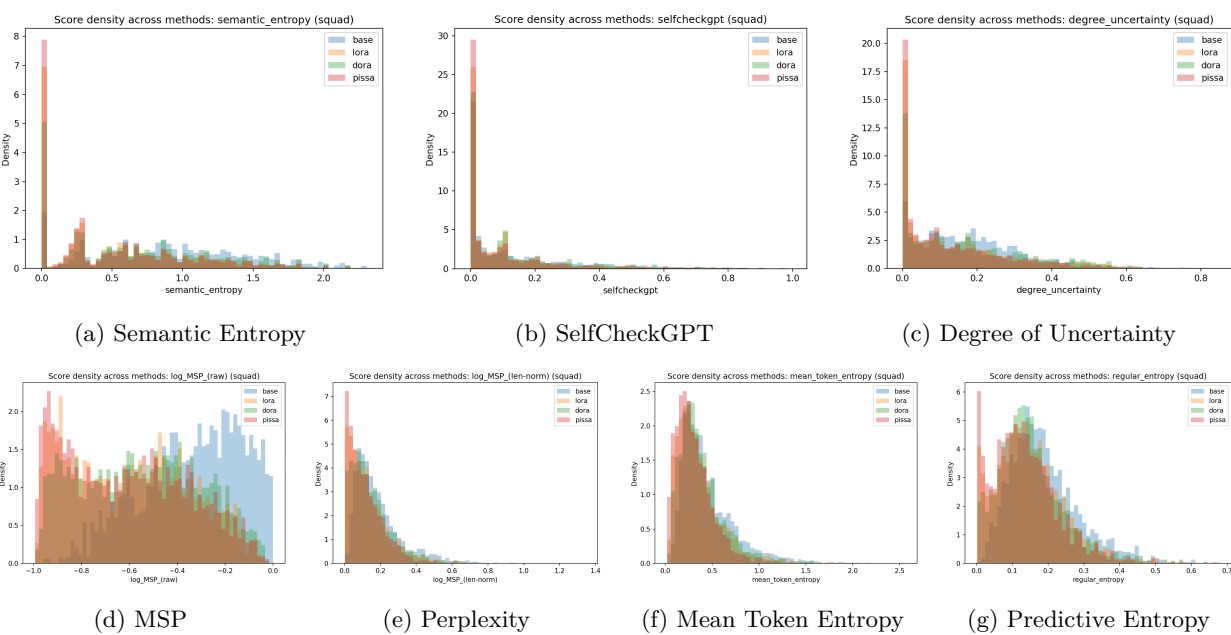

(a) Semantic Entropy      (b) SelfCheckGPT      (c) Degree of Uncertainty

(d) MSP      (e) Perplexity      (f) Mean Token Entropy      (g) Predictive Entropy

Figure 8: Uncertainty score density distributions across PEFT methods on LLaMA-3.2-3B-Instruct (SQuAD). Top row: semantic-level detectors. Bottom row: token-level detectors. X-axis represents the uncertainty score.

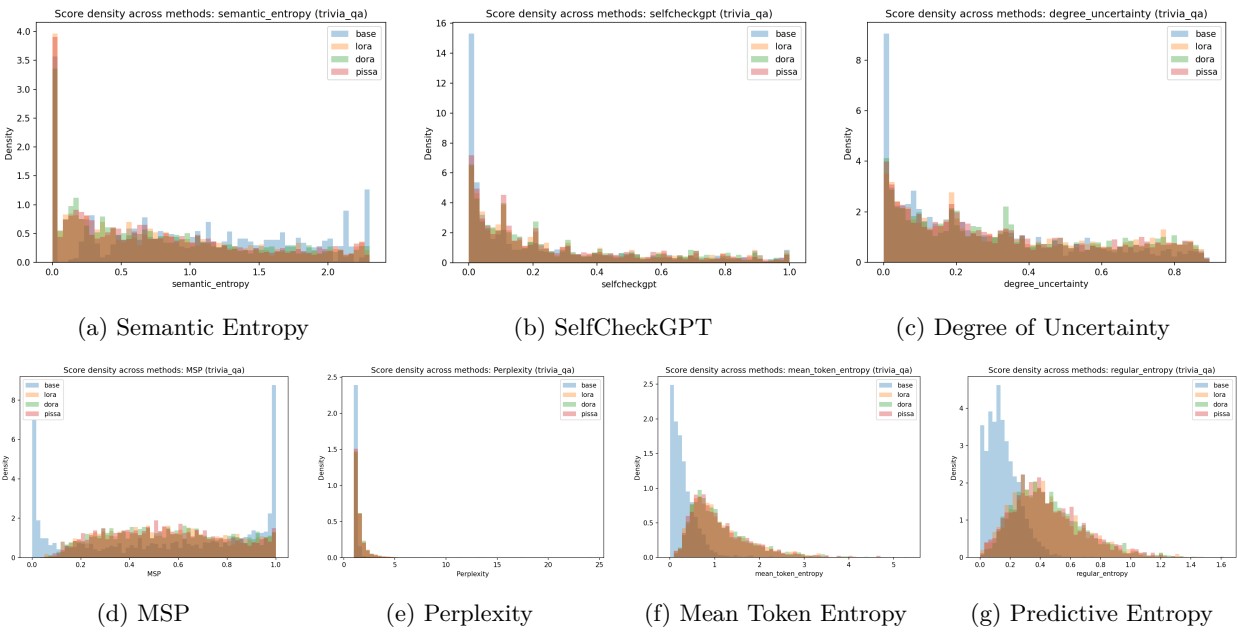

(a) Semantic Entropy   (b) SelfCheckGPT   (c) Degree of Uncertainty

(d) MSP   (e) Perplexity   (f) Mean Token Entropy   (g) Predictive Entropy

Figure 9: Uncertainty score density distributions across PEFT methods on Mistral-7B-Instruct-v0.3 (Trivia-aQA). Top row: semantic-level detectors. Bottom row: token-level detectors. X-axis represents the uncertainty score.

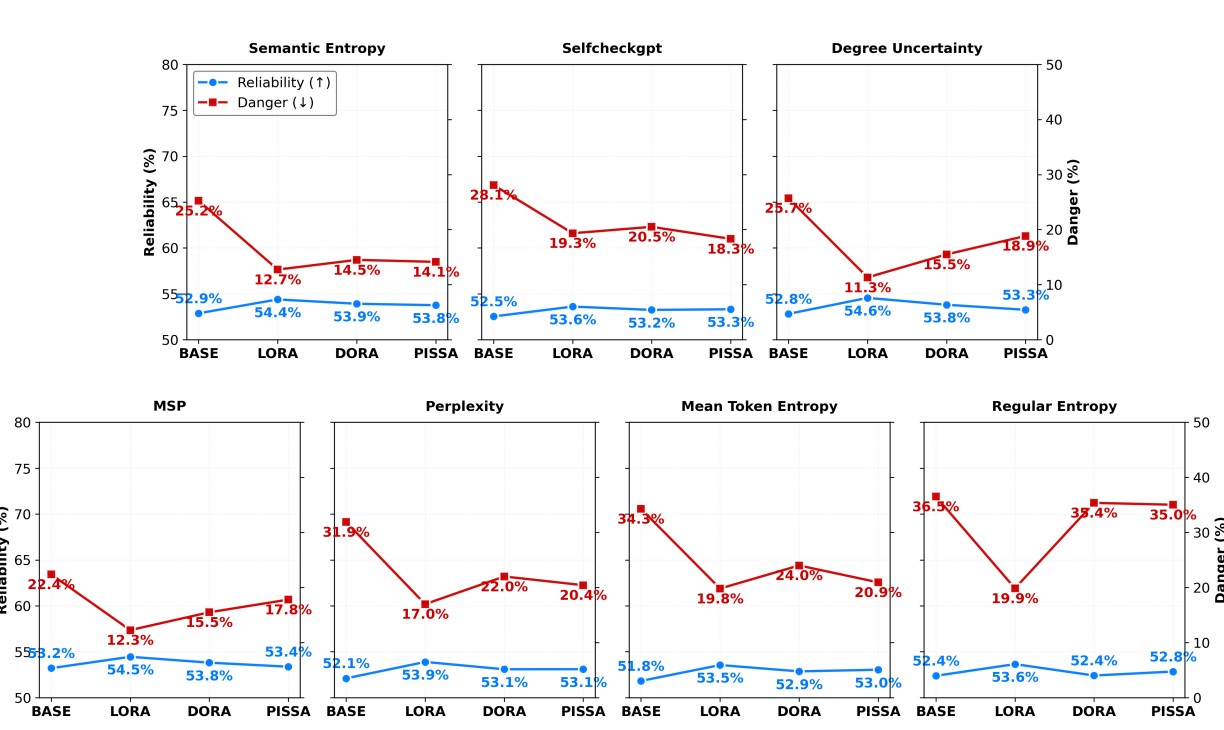

Figure 10: Confidence-correctness safety analysis on SQuAD (Llama-3.2-3B). Reliability (↑): precision of high-confidence predictions. Danger (↓): fraction of errors delivered with high confidence. PEFT methods reduce dangerous hallucinations while maintaining prediction reliability.

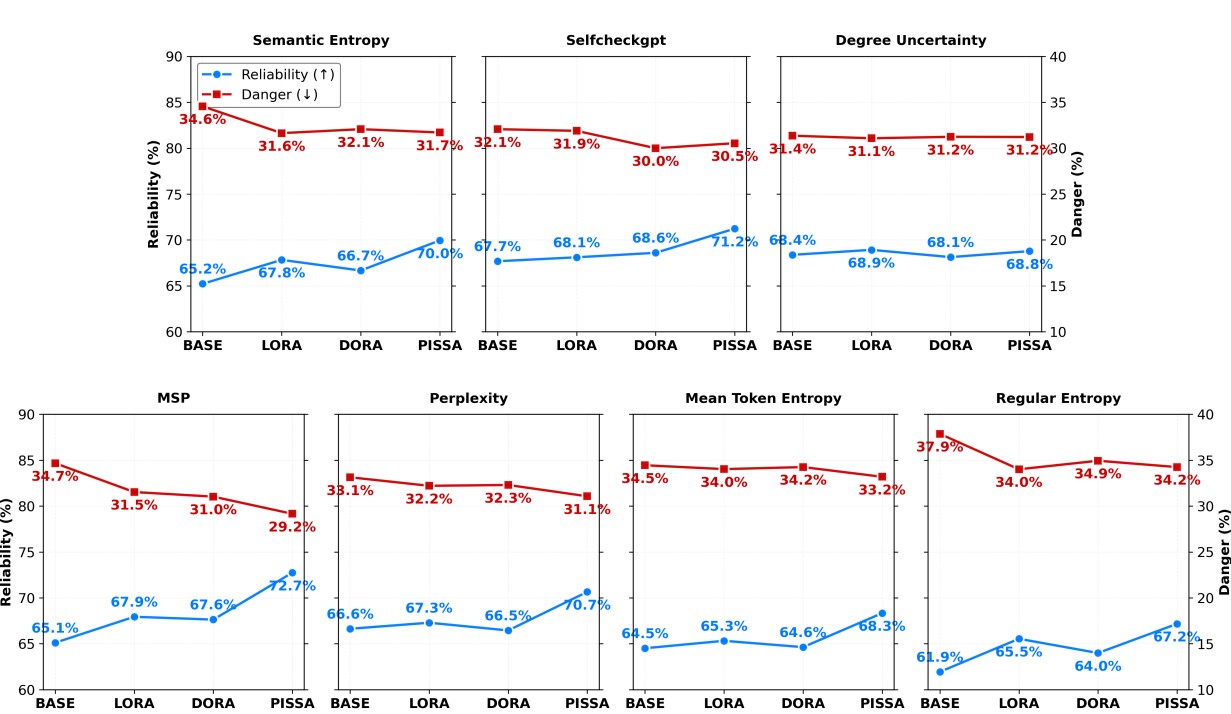

Figure 11: Confidence-correctness safety analysis on NQ-Open (Llama-3.2-3B). Reliability (↑): precision of high-confidence predictions. Danger (↓): fraction of errors delivered with high confidence. PEFT methods reduce dangerous hallucinations while maintaining prediction reliability.

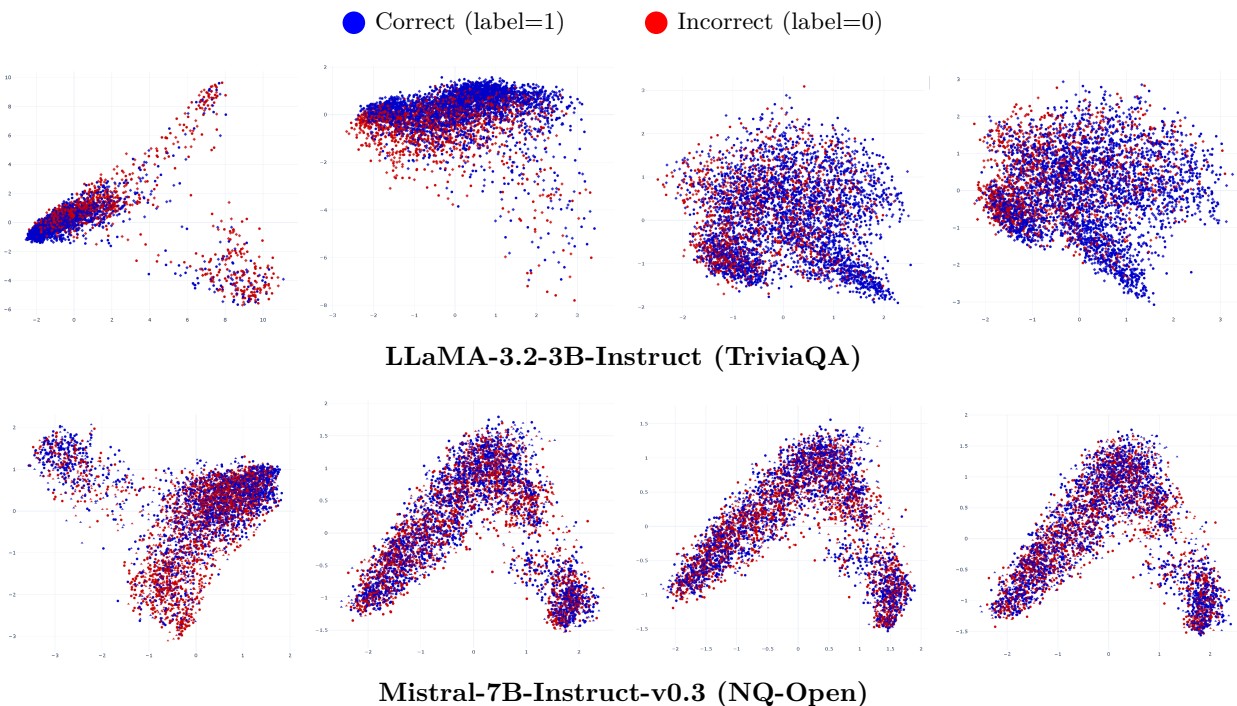

Figure 12: PCA visualization of hidden representations at the best probing layer. Each row shows Base, LoRA, DoRA, and PiSSA (left to right). It is obvious that PEFT methods almost do not separate the labels clearly.

