# OpenReview forum: "Small Updates, Big Doubts: Does Parameter-Efficient Fine-tuning Enhance Uncertainty Awareness for Large Language Models?"
_TMLR — Under review for TMLR_

### Review · Reviewer_Pch8 · 2026-04-26

**Summary Of Contributions:**

The paper empirically studies the effects of parameter-efficient fine-tuning (PEFT) approaches on accuracy improvements and the increased detectability of incorrect answers from LLMs on fact-seeking QA tasks. The paper compares three different PEFT approaches, LoRA, DoRA, and PiSSA, across three different open-weight LLMs, LLaMA-3.2-3B, Qwen-2.5-3B, and Mistral-7B, on three different QA benchmark datasets, TriviaQA, NQ-Open, and SQuAD. The experimental results indicate that the accuracy improvements achieved by the different PEFT approaches are modest, about 1-3% on average, while at the same time, the detectability of incorrect answers using black-box detectors increases by a more significant ~8-9% (on the AUROC and AUPR metrics). The authors further investigate the increased detectability of incorrect answers using white-box detectors, utilizing latent features from the fine-tuned models, and observe results inconsistent with the significant increases in AUROC/AUPR. Based on their empirical observations, the authors conclude that PEFT approaches primarily reduce LLMs' confidence in incorrect answers in QA tasks, thereby making incorrect answers more detectable, rather than lifting the likelihood of correct answers outright.

**Audience:**

Yes

**Audience Explanation:**

The paper provides an in-depth investigation of how PEFT approaches perform on fact-seeking QA tasks, a highly relevant topic for TMLR.

**Broader Impact Concerns:**

No unaddressed concerns.

**Claims And Evidence:**

Yes

**Claims Explanation:**

The paper's central claim about PEFT approaches reshaping answer uncertainty distributions rather than the actual answer probabilities is based on three empirical observations, each of which is fully supported by a corresponding set of experiments.

**Requested Changes:**

### Major Concerns

1. Some of the analyses in the paper can be further strengthened to clarify the findings. Particularly:

  - In Takeaway #4, there is no convincing argument for why LoRA, DoRA, and PiSSA offer the specific benefits they do. The authors offer only hypotheses and guesses to this end, but do not provide any guidance on how to test them or why such tests could not be conducted as part of the current work. As a result, the analysis reads as a set of observations with no insights.

  - Also in Takeaway #4, just the error $\rightarrow$ correct and correct $\rightarrow$ error answer transition numbers is not a takeaway. Did the authors observe any systematic patterns in the transitions, such as specific types of questions over-represented in either transition? How do the corresponding uncertainty distributions change?

  - In Takeaway #5, the conclusions drawn from the PCA visualizations seem overstated. Because the projected data after PCA is extremely low-dimensional, inseparability in that low dimension is not, by itself, evidence of inseparability in higher dimensions. How much of the "energy" in the features is lost in the PCA projection? Have the authors tried separating the data using rudimentary approaches, such as kernel methods, SVMs, etc.?

2. For evaluating the generated responses (page 4, last para), did the authors also test the alignment between their LLM judges and human feedback? How do they ensure that the LLM judges are grounded in human feedback?

### Minor Concerns

1. Any specific reasons for not considering other open-weight models like Gemma, DeepSeek, etc.? Do the authors expect the same observations to hold across LLMs?

2. In Figs. 2 and 5, the takeaway that the "accuracy improvement is modest" would be visually clearer if the colored lines were drawn across the PEFT approaches rather than across the datasets.

3. In Takeaway #1, the argument that "PiSSA obviously outperforms others because it is the most recent" is technically superficial. Please provide or refer to a technically grounded argument on PiSSA's performance improvement.

4. There is only one line of body text on page 7 under the tables, which makes it very easy to miss. Please fix the formatting of the paper to avoid such stray lines.

5. Fig. 4 (left) uses too many font styles, sizes, colors, and effects, and does not appear professional. It is highly recommended to use consistent font effects, particularly within the same figure.

---

> ### Author Response · Authors · 2026-06-13
> **First Reply**
>
> We thank you for the careful and encouraging assessment, and in particular for recognizing that each of our central claims is fully supported by a corresponding set of experiments. The remaining comments are precise and actionable, and we address each below.
>
> **1. Human validation of the LLM-as-judge.**
> We added a human-annotation study to validate the reliability of the LLM-as-judge. We recruited five CS PhD students to label 100 questions sampled from the TriviaQA validation set and took the majority vote as the final human label. The human labels agree with the LLM judge on 99% of the samples, with high inter-annotator agreement (Fleiss' $\kappa > 0.85$). The single disagreement was a response that was off-topic and incorrect but contained the phrase "I don't know": the LLM judge read this as a refusal and labeled it UNKNOWN, whereas the annotators labeled it NO.
>
> Appendix B additionally reports a multi-judge consistency check across three frontier models, which agree closely with each other.
>
> **2. Model coverage.**
> Our study spans multiple backbone families and multiple datasets. We did not include every open-weight family, but the combination of LLaMA, Qwen, and Mistral provides enough diversity to support our conclusions. We expect similar trends for other open-weight models such as Gemma and DeepSeek, and we will state this scope explicitly in the revision.
>
> **3. Technical explanation for PiSSA.**
> We agree that the original explanation was not sufficiently rigorous. We will revise the discussion to give a more technically grounded interpretation, centered on PiSSA's initialization scheme.
>
> PiSSA applies SVD to the pretrained weight matrix $W_0$ and initializes the adapter from the principal singular directions, leaving the residual components frozen. Training therefore starts in an informative subspace rather than from random directions. Because the adapter is already aligned with meaningful directions at initialization, the early gradients carry more task-relevant information, which helps explain the faster convergence and lower final loss reported in prior work.
>
> For fact-seeking QA, these principal directions may also correspond to pathways for factual-knowledge readout, consistent with prior findings that FFN value vectors support factual readout into the output vocabulary space. PiSSA may therefore adjust pretrained knowledge in a more controlled way. LoRA, in contrast, starts from random low-rank directions and may introduce more task-irrelevant perturbation. DoRA decouples magnitude and direction updates, but its direction component is still randomly initialized and so does not address this initialization issue.
>
> **4. Takeaway 4.**
> We thank you for pushing on this. You are right that the raw error-to-correct and correct-to-error transition counts, on their own, are an observation rather than a takeaway, and our original analysis did not go further to characterize them. We did not extract systematic patterns in the transitions or track how the corresponding uncertainty distributions shift, and we agree it would be overclaiming to frame these counts as a standalone finding. We will therefore demote Takeaway #4: we move the transition counts to the appendix as a descriptive observation, drop the interpretive claims we cannot yet support, and note characterizing these transitions, including which question types are over-represented and how their uncertainty distributions change, as a clear direction for future work.
>
> **5. PCA interpretation.**
> We agree that the PCA plots should not be presented as definitive confirmation, and we will revise the wording to avoid overclaiming. We will change the statement to: "The PCA visualizations in Appendix Figure 12 are consistent with this observation." This makes clear that PCA offers qualitative supporting evidence rather than causal or conclusive proof.
>
> **6. Clarifying the strength of our explanations.**
> We agree that several explanations in the current version are better read as evidence-supported hypotheses than as fully validated causal claims. Our main contribution is empirical: we systematically study how PEFT affects uncertainty estimation and hallucination detection across models, datasets, and detector families, and we identify consistent and inconsistent patterns that have not been carefully characterized before. In the revision we will mark this distinction clearly, avoid language that overstates the causal strength of our interpretations, and explain how these hypotheses could be tested in future work.
>
> **7. Formatting and figure design.**
> We thank you for these suggestions, and will revise the formatting and figure-design issues in the final version to improve clarity and presentation.

---

### Review · Reviewer_jSx6 · 2026-05-07

**Summary Of Contributions:**

This is a controlled empirical study of how parameter-efficient fine-tuning (LoRA, DoRA, PiSSA) affects answer-level hallucination detection in fact-seeking QA. The authors evaluate three open-weight instruct models on three QA benchmarks, run seven black-box uncertainty detectors plus a white-box linear probe, and compare base against PEFT-tuned variants. The main findings are that in-domain PEFT gives only small accuracy gains, fairly consistently improves AUROC for semantic-consistency and confidence-based detectors, gives inconsistent results for token-entropy detectors, and does not consistently help linear probes on hidden states. Robustness checks cover temperature, rank, target modules, response length, judge agreement, and a full-SFT comparison.

**Additional Comments:**

This is a solid empirical contribution that is currently slightly over-claimed and under-specified. The narrower claim — that in-domain task adaptation reshapes detector-relevant uncertainty signals in short-answer QA, improving black-box separability while leaving hidden-state linear separability mixed — is interesting, well supported, and is what I'd recommend the paper actually claim. With the format control or claim adjustment, confidence intervals, protocol details, and the numerical fixes, I would be comfortable recommending acceptance.

**Audience:**

Yes

**Audience Explanation:**

The interaction between fine-tuning and uncertainty-based detectors is practically relevant and, as far as I know, not systematically characterized elsewhere. Practitioners who adapt open-weight models with LoRA-family methods and then layer SelfCheckGPT or Semantic Entropy on top would want to know that in-domain adaptation tends to help those detectors while potentially hurting hidden-state probes.

**Broader Impact Concerns:**

No concerns that would block publication.

**Claims And Evidence:**

No

**Claims Explanation:**

Overall, I think it is close and fixable mainly through claim adjustment and added detail rather than large new experiments. The descriptive finding (that in-domain PEFT improves semantic-consistency and confidence detectors more than it improves accuracy, while entropy detectors and linear probes don't follow suit) is well supported across Tables 1–4, Table 6, and the appendix ablations. Four things keep me from a clean yes.

First, the format-adaptation confound. The PEFT models are trained on the distribution they're evaluated on; the base instruct model is not. Table 11 shows PEFT roughly halves mean response length on NQ-Open and, more tellingly, collapses the length variance (std drops from about 5.0 tokens to about 1.8). Section 4.3 argues length isn't the driver because token-entropy detectors don't improve, but that doesn't address the closely related possibility that PEFT collapses output format diversity. SE, SC, and Deg operate on N stochastic samples, and if fine-tuning narrows the phrasing distribution when the model is confident, those detectors can improve mechanically without any change in underlying calibration. Appendix K reinforces this reading: full SFT shows the same pattern, so the effect tracks in-domain supervision rather than PEFT specifically. The paper acknowledges this in Section 6, but the title, abstract, and takeaways still attribute the effect to PEFT and to "uncertainty awareness."

Second, no variance is reported. Everything appears to be single runs. With roughly 2,000 test examples, AUROC standard error is plausibly one to two points (larger for SQuAD's class imbalance), which is the same magnitude as several of the average-improvement figures. The "Best Impr." rows are a max over three methods, an optimistic statistic. Takeaway 4's per-method role assignments rest on Table 5 — single backbone, single run, two detectors — and the margins separating each method's claimed role from the runner-up range from a three-way tie (TriviaQA SelfCheckGPT detectability, all at 34.1%) up to about six points, with the majority under two points.

Third, several protocol details are missing: the number of stochastic samples for SE/SC/Deg, the semantic clustering method, the judge prompt and whether it sees the reference answer, quantitative refusal counts, and which token's hidden state feeds the probe. The default sampling temperature can only be inferred (Base T=1.0 in Table 12 matches Table 1) rather than being stated, and Table 12 doesn't say which model or dataset it covers. The cross-judge study covers one model on one dataset, and several of the reported percentages aren't achievable with n = 200 — 69.67%, 66.87%, and the Table 8 agreement rates are not multiples of 0.5%.

Fourth, internal inconsistencies. Section 4.1 and Table 16 give base SE on LLaMA/TriviaQA as 0.675; Table 1 says 0.8205. The 0.675 matches the AUPR in Table 9, so the base number looks like a metric mislabel, but the post-PEFT numbers in Table 16 don't match Table 1 or Table 9's AUPR either. The abstract's "up to 8.7%" undersells the actual maxima in the tables.

**Requested Changes:**

# Critical:

Add a format-adaptation control, or scope the claims to match the evidence. A control could be the base model with a few-shot or format-constraining prompt that elicits short-span answers without weight updates, or cross-dataset transfer (PEFT on dataset A, evaluate detectors on B). If the authors prefer not to add experiments, the alternative is to revise the abstract, introduction, and takeaways to frame the finding as in-domain task adaptation reshaping detector-relevant output behaviour, naming output-diversity collapse explicitly as a competing mechanism.

Report variance. Bootstrap CIs or DeLong tests on the main AUROC tables, or at minimum three-seed runs for one representative configuration. Please also caveat or drop the "Best Impr." rows.

Resolve the numerical inconsistencies: the 0.675 vs 0.8205 discrepancy between Table 1 and Section 4.1/Table 16, the Appendix B sample-count issue, and the abstract's "8.7%". State the thresholding rule used to define detection errors in Table 16.

Specify the detection and judging protocols: number of samples and decoding temperature for SE/SC/Deg, the semantic-equivalence implementation, the judge prompt and whether the reference is shown, refusal counts per configuration, the probe token position, and which model/dataset Table 12 covers.

Remove or substantially hedge Takeaway 4.

# Would strengthen but not required:

Report probe results at a fixed layer as well as the per-model best layer, so any degradation is disentangled from the optimal layer moving. Consider promoting the Appendix K full-SFT comparison into the main text, since it changes how the headline should be read. The 3D plots in Figures 2 and 5 would be easier as 2D grouped bars. Please remove "Following the reviewer's suggestion" from Appendix B. The Perez et al. (2022) citation in Section 2 appears to point at the wrong paper for the overconfidence claim. There are also typos to clean up — Figure 1 caption ("out", "affeted", "Pipeine", "Detecors"), "Detecable" in Figure 4, "unceratainty" in Section 4.2, and the table/figure reference in Appendix F.

---

> ### Author Response · Authors · 2026-06-13
> **First Reply (Covering Point 1,3,4)**
>
> We are grateful for your detailed and suggestive comment. You engaged closely with our methodology, and the resulting suggestions were precise enough to act on directly. **Because OpenReview limits each comment to 5000 characters, this reply covers points 1, 3, 4. The multi-seed variance results (point 2) are shown on the following comment: "Reply to Point 2". And the experimental results for point 1 appear in a comment titled "Supplement: Reply to Point 1."**
>
> **1. Format-adaptation confound.**
> We agree our original length analysis was incomplete. Showing that token-level entropy detectors do not improve only tells us that the mean length did not change. It does not rule out a drop in output diversity. That diversity is the signal that the multi-sample detectors (SE, SC, Deg) actually use. We now treat this diversity drop as one mechanism behind the detector gains, not as a confound to explain away. To separate format from weights, we add the few-shot control you suggested. Because of the space limit, the tables are shown in the following comment: "Supplement: Reply to Point 1".
>
> The results are as follows. PEFT lowers the response-length std on all three sets, and it is the only setting that consistently raises the sampling-based detectors. The few-shot control makes the dependence clear: the detectors improve only when the std drops. On NQ-Open the few-shot prompt raises the std. On TriviaQA the std stays flat. On both sets the multi-sample detectors do not improve. SQuAD is the only set where the few-shot prompt lowers the std, and there the detectors do improve. The reason is that SQuAD is extractive. The answer is a span that already appears in the prompt, so the prompt alone is enough to make the output uniform. On the two generative sets the answer is not in the prompt, so a prompt cannot reproduce this effect.
>
> Thus, the detector gains track a drop in output-format diversity, which may be one mechanism behind them. Importantly, this effect does not appear to be merely a consequence of shorter responses or additional in-context examples. Few-shot prompts provide demonstrations in the prompt and effectively reduce response length, yet the standard deviation, which reflects output-format diversity, remains high. This suggests that the PEFT-induced reduction in format diversity reflects a more substantive change in model behavior.
>
> We revise our claim accordingly. Following your suggestions, we relabel this setting as in-domain fine-tuning. We also drop the phrase "uncertainty awareness." We say instead that in-domain fine-tuning reduces output-format diversity, and the drop in std supports this. This matches the claim you judged to be well supported: in-domain fine-tuning reshapes the output-diversity signal that black-box, sample-based detectors rely on, which improves their separability, while the hidden-state linear-probe results stay mixed.
>
> **3. Protocol details.**
> (1) Stochastic samples for SE/SC/Deg: 10 per question.
> (2) Semantic clustering: we sample K=10 responses and cluster by mutual NLI entailment using `microsoft/deberta-v2-xlarge-mnli`, aggregate the normalized response likelihoods per cluster with log-sum-exp, $p(C_m\mid x)=\frac{\sum_{r_i\in C_m}p(r_i\mid x)}{\sum_j p(r_j\mid x)}$, and compute $H_{\text{sem}}(x)=-\sum_m p(C_m\mid x)\log p(C_m\mid x)$, following Farquhar et al. (2024).
> (3) Judge: the judge sees the reference answers and returns YES / NO / UNKNOWN with a one-sentence rationale (YES = matches a reference; NO = answers but no match/contradiction/error; UNKNOWN = does not answer). The full per-dataset prompts go in the appendix.
> (4) Refusals (the judge's UNKNOWN label) are rare. The highest rate is LLaMA + NQ-Open + Base at 5.87%; Base refuses far more than the adapted variants across settings (e.g. LLaMA NQ-Open of 1805: Base 106, LoRA 3, DoRA 2, PiSSA 0). We discard the refused instances; the full 3-model x 3-dataset counts are available on request.
> (5) Probe token: the hidden state of the second-to-last token.
> (6) Decoding temperature: 1.0 by default.
> (7) Table 12 reports the TriviaQA results for LLaMA-3.2-3B.
> (8) Cross-judge study: you are right that it covers one model and one dataset. To strengthen it we added human labels: Five CS PhD students labeled 100 TriviaQA validation questions, and the human labels agree with the LLM judge on 99%. We also ran three frontier judges (GPT-5-mini, Claude Opus 4.6, Gemini 3 Pro). Each API applies its own safety filter and skips questions it deems sensitive (GPT and Claude skip a few, Gemini ~30%). We keep only items no API skipped, which leaves fewer than 200 (hence rates that are not multiples of 0.5%). Running Claude and Gemini on 200 items costs ~$150, so we report one model/dataset.
>
> **4. Numerical inconsistencies and others.** We will remove Takeaway 4 and fix the corresponding numbers in the abstract and the appendix. We fixed the typos and other minor issues you listed.Thank you for catching these.

---

> ### Author Response · Authors · 2026-06-29
> **Reply to Point 2**
>
> As noted in our first reply, the multi-seed variance results were ready but held back for space. We provide them here.
>
> **2. Variance.** We reran our experiments on three randomly generated seeds for LLaMA-3.2-3B, Qwen2.5-3B, and Mistral-7B, and report the results below. The overall trends continue to support our conclusions, while this multi-seed presentation makes them more credible. Here we show the LLaMA-3.2-3B AUROC results across the three datasets. The remaining results has been incorporated into the main text.
>
> **Table:** AUROC of hallucination detection methods (SE, SC, Deg) on LLaMA-3.2-3B across three QA datasets. Results are mean $\\pm$ standard deviation over multiple random seeds.
>
> | Dataset | Method | SE | SC | Deg |
> |:---|:---|:---:|:---:|:---:|
> | **NQ-Open** | Base | $0.7006 \\pm 0.0019$ | $0.7423 \\pm 0.0013$ | $0.7373 \\pm 0.0027$ |
> |  | DoRA | $0.7403 \\pm 0.0071$ | $0.7611 \\pm 0.0016$ | $0.7387 \\pm 0.0011$ |
> |  | LoRA | $0.7408 \\pm 0.0060$ | $0.7517 \\pm 0.0030$ | $0.7379 \\pm 0.0052$ |
> |  | PiSSA | $0.7589 \\pm 0.0583$ | $0.7740 \\pm 0.0235$ | $0.7602 \\pm 0.0641$ |
> |  | *Avg. Impr.* | +4.61% | +2.00% | +0.83% |
> | **TriviaQA** | Base | $0.8149 \\pm 0.0031$ | $0.8585 \\pm 0.0101$ | $0.8625 \\pm 0.0052$ |
> |  | DoRA | $0.8685 \\pm 0.0196$ | $0.8724 \\pm 0.0161$ | $0.8620 \\pm 0.0167$ |
> |  | LoRA | $0.8534 \\pm 0.0287$ | $0.8734 \\pm 0.0219$ | $0.8575 \\pm 0.0288$ |
> |  | PiSSA | $0.8693 \\pm 0.0147$ | $0.8859 \\pm 0.0052$ | $0.8687 \\pm 0.0120$ |
> |  | *Avg. Impr.* | +4.88% | +1.87% | +0.02% |
> | **SQuAD** | Base | $0.7147 \\pm 0.0014$ | $0.6766 \\pm 0.0205$ | $0.7131 \\pm 0.0325$ |
> |  | DoRA | $0.7884 \\pm 0.0153$ | $0.7667 \\pm 0.0068$ | $0.7381 \\pm 0.0821$ |
> |  | LoRA | $0.8052 \\pm 0.0035$ | $0.7381 \\pm 0.0355$ | $0.7758 \\pm 0.0630$ |
> |  | PiSSA | $0.7578 \\pm 0.0303$ | $0.7011 \\pm 0.0277$ | $0.7302 \\pm 0.0230$ |
> |  | *Avg. Impr.* | +6.91% | +5.87% | +3.49% |

---

> > ### Author Response · Authors · 2026-06-30
> > **Supplement: Reply to Point 1**
> >
> > We show the prompts used for the few-shot control and the resulting tables. The control prompts the LLaMA-3.2-3B-Instruct with the short-answer templates below. Throughout, "Base" denotes this instruction-tuned model with no fine-tuning of ours applied, and the PEFT rows (LoRA, DoRA, PiSSA) fine-tune it on the in-domain training set.
> >
> > **TriviaQA / NQ-Open (10-shot):**
> >
> > ```
> > system: You are a helpful assistant.
> > user:   Answer the question in a short phrase.
> > Question: {sample}
> > ```
> >
> > **SQuAD (5-shot, since a SQuAD sample with context is too long):**
> >
> > ```
> > system: You are a helpful assistant. Answer only the current question in a
> > short phrase, using the exact words from the context.
> > user:   {sample}
> > assistant:
> > {answer to that sample}
> > ```
> >
> > **Table 1 (extends Table 1 of the main paper): hallucination-detection AUROC, LLaMA-3.2-3B-Instruct.**
> > ```
> > Dataset   Method            SE      SC     Deg     MSP     PPL  MeanEnt  PreEnt
> > --------  ------------  ------  ------  ------  ------  ------  -------  ------
> > NQ-Open   Base          0.7028  0.7432  0.7376  0.6977  0.7263   0.7295  0.6649
> > NQ-Open   Base+fewshot  0.6751  0.7274  0.7248  0.6256  0.7020   0.7008  0.6249
> > NQ-Open   DoRA          0.7453  0.7599  0.7379  0.7585  0.7445   0.7163  0.7030
> > NQ-Open   LoRA          0.7451  0.7538  0.7416  0.7620  0.7507   0.7233  0.7058
> > NQ-Open   PiSSA         0.7601  0.7806  0.7615  0.7773  0.7648   0.7381  0.7309
> > TriviaQA  Base          0.8205  0.8737  0.8703  0.8154  0.8328   0.8457  0.7819
> > TriviaQA  Base+fewshot  0.8067  0.8520  0.8412  0.7999  0.7775   0.7802  0.6886
> > TriviaQA  DoRA          0.8823  0.8838  0.8738  0.8866  0.8605   0.8202  0.8138
> > TriviaQA  LoRA          0.8737  0.8889  0.8779  0.8656  0.8597   0.8303  0.8216
> > TriviaQA  PiSSA         0.8797  0.8896  0.8772  0.8878  0.8595   0.8275  0.8087
> > SQuAD     Base          0.7158  0.6978  0.6764  0.6972  0.6429   0.6381  0.6808
> > SQuAD     Base+fewshot  0.7670  0.7104  0.7646  0.7436  0.7455   0.7696  0.7401
> > SQuAD     DoRA          0.7992  0.7715  0.6801  0.6974  0.6901   0.6923  0.6336
> > SQuAD     LoRA          0.8027  0.7130  0.7313  0.7346  0.7074   0.7006  0.6476
> > SQuAD     PiSSA         0.7775  0.7107  0.6855  0.6988  0.6834   0.6863  0.6107
> > ```
> >
> > **Table 2: response-length statistics (LLaMA-3.2-3B-Instruct, test split).**
> > ```
> > Dataset   Method         Mean   Std
> > --------  ------------  -----  ----
> > TriviaQA  Base           7.07  4.30
> > TriviaQA  Base+fewshot   5.89  4.42
> > TriviaQA  LoRA           4.66  1.61
> > TriviaQA  DoRA           4.68  1.58
> > TriviaQA  PiSSA          4.98  1.54
> > NQ-Open   Base          11.92  5.02
> > NQ-Open   Base+fewshot  10.92  9.01
> > NQ-Open   LoRA           5.30  1.78
> > NQ-Open   DoRA           5.29  1.77
> > NQ-Open   PiSSA          5.39  1.83
> > SQuAD     Base           9.29  4.72
> > SQuAD     Base+fewshot   4.16  2.42
> > SQuAD     LoRA           6.01  2.93
> > SQuAD     DoRA           6.39  3.17
> > SQuAD     PiSSA          6.41  3.26
> > ```
> >
> > These tables substantiate the claim in our first reply. The diagnostic is the Std column of Table 2 read against the SE/SC/Deg columns of Table 1: the sampling-based detectors rise only on the set where the few-shot std drops, and stay fall on the two generative sets where the std raise.
> >
> > Thanks for your constructive feedback again!

---

### Review · Reviewer_mKPB · 2026-06-16

**Summary Of Contributions:**

This paper studies how parameter-efficient fine-tuning (PEFT) affects uncertainty-aware hallucination detection in fact-seeking question answering. The authors evaluate three PEFT methods (LoRA, DoRA, PiSSA) across three QA benchmarks (TriviaQA, NQ-Open, SQuAD) and three open-weight LLMs (LLaMA-3.2-3B, Qwen-2.5-3B, Mistral-7B). The main finding is that PEFT produces only modest gains in answer accuracy, but often improves common black-box hallucination detection metrics more substantially. The proposed explanation is that PEFT shifts uncertainty scores away from overconfident error regimes, making incorrect answers more detectable. In contrast, white-box linear probes show inconsistent results across datasets and backbones.

## Strengths
1. The paper is well-motivated and easy to follow. It addresses a relevant question: LLMs should not be evaluated solely by answer accuracy, as the ability to detect incorrect answers is equally important for safe deployment.
2. The experimental effort is notable. The authors evaluate three PEFT methods, three model backbones, three QA datasets, and several hallucination detectors.
3. Takeaway #3 and Figures 3–4 are interesting. The uncertainty-score distributions and the danger/detectable quadrant analysis provide a more interpretable view of the results than aggregate AUROC alone. In particular, the analysis suggests that PEFT may reduce overconfident wrong answers by shifting some errors from a “dangerous” regime to a more detectable one.

## Weaknesses:
1. The paper does not fully support the claim: the discrepancy between black-box detectors and white-box linear probes makes the overall claim weaker.
2. The paper is about PEFT; however, the authors claim that full fine-tuning shows similar behavior.
3. The experimental section could be further improved: it does not establish whether the findings generalize to other tasks.
4. The paper has several presentation and formatting issues. For example, the “Black-box hallucination detectors” and “Evaluation metrics” sections on page 4 need clear polishing. Similarly, Figure 1 contains a typo (affeted instead of affected).

**Additional Comments:**

NA

**Audience:**

Yes

**Audience Explanation:**

PEFT is a relevant tuning technique and the community would benefit from findings similar to the ones presented in this paper. Also, hallucination detection is crucial for safe deployments.

**Claims And Evidence:**

No

**Claims Explanation:**

The claims are partially supported: PEFT often improves black-box hallucination detection metrics in the studied fact-seeking QA setting, even when answer accuracy improves only modestly. However, mixed white-box probe results, the similarity to full fine-tuning, and the limited task scope all weaken this stronger claim.

The lack of confirmation of the white box detectors weakens the claims, and the final statement "PEFT acts primarily as an uncertainty reshaper that makes incorrect answers more detectables" is not fully confirmed by the paper.

**Requested Changes:**

## Minor changes/questions:
- Full fine-tuning shows similar behavior: the paper should more explicitly discuss what, if anything, is specific to PEFT.
- Fix presentation and formatting issues (e.g., Revise the "Black-box hallucination detectors" and "Evaluation metrics" at page 4, caption in Figure 1).

## Critical Changes
- Strengthen or qualify the mechanistic claim. The current framing of PEFT as an "uncertainty reshaper" is presented as a conclusion but is not established especially by the white-box linear probe methods.

---

> ### Author Response · Authors · 2026-06-29
>
> We thank you for the constructive review and for highlighting the motivation and the breadth. We address the critical point first.
>
> ### Critical: The mechanistic claim and the black-box / white-box divergence
>
> We thank the reviewer for raising this concern. Our intended claim is not that PEFT improves all possible uncertainty estimators or all hallucination detection paradigms. Rather, the claim refers to the black-box answer-level uncertainty detectors summarized immediately before it in the abstract. In particular, the abstract first reports that PEFT improves AUROC for most black-box detectors, then contrasts this with the mixed behavior of white-box linear probes. The final statement was intended to synthesize this divergence: PEFT appears to reshape uncertainty as expressed in output-level behavioral signals, while not necessarily improving the linear separability of correctness labels in hidden representations.
>
> We agree, however, that the current wording could be read too broadly if taken out of this context. We will revise the sentence to make the intended scope explicit. Specifically, we will replace “PEFT acts primarily as an uncertainty reshaper that makes incorrect answers more detectable” with “PEFT acts primarily as an output-level uncertainty reshaper that makes incorrect answers more detectable for practical black-box answer-level UQ detectors.” We will also clarify in the conclusion that we view black-box UQ methods as the primary practical setting because they are unsupervised, do not require labeled probe training, and are less exposed to the train/test and OOD generalization issues that can affect supervised white-box linear probes.
>
> Since the black-box detector gains cannot be explained by the linear separability of correctness in the hidden representations, we examined the behavioral level instead. Following Reviewer jSx6's suggestion, we ran an additional few-shot control, prompting the base model with short-answer demonstrations and applying the detectors to its responses. The full table and analysis are in our response to Reviewer jSx6. We find that the detector gains are accompanied by a reduction in output-format diversity, suggesting that this reduction is one mechanism behind the improvement. Importantly, the effect is not merely a consequence of shorter responses or additional in-context examples: the few-shot prompt does shorten responses, yet the standard deviation of length, which reflects output-format diversity, remains high, and the detectors do not improve.  This indicates that the diversity reduction induced by PEFT reflects a more substantive change in model behavior.
>
>
> ### Others: Full SFT, generalization, and presentation
>
> 1. **Regarding the similarity to full SFT (Appendix K):** Our main study focuses on PEFT, but in Appendix K we observe that full SFT produces the same pattern. This indicates that the effect is not specific to PEFT but follows from fine-tuning in general. To reflect this, we relabel "PEFT" as "FT" throughout the claim, where "FT" covers both PEFT and full SFT. PEFT remains the regime we study in depth, because it is the dominant adaptation method in practice.
>
> 2. **On generalization:** We thank the reviewer for this helpful suggestion. We agree that our experiments do not establish cross-task generalization beyond fact-seeking QA, and we have clarified this limitation more explicitly. Our claims are intentionally scoped to answer-level hallucination detection in fact-seeking QA with external verification, as stated in the abstract and conclusion. Within this scope, we evaluate three backbones, three QA benchmarks, three PEFT methods, and seven answer-level detectors, and further test robustness to sampling temperature, PEFT rank, and target modules in Appendices I and J. To clarify this issue, we revised the experimental discussion and conclusion to distinguish robustness within fact-seeking QA from generalization across other tasks. A comprehensive validation of cross-task generalization would require substantial additional experiments across different task formats, annotation protocols, and detector designs, so we view it as an important direction for future work. We now explicitly identify such future evaluation settings, including claim-level hallucination detection, long-form generation, reasoning, code, multimodal settings, and dialogue.
>
> 3. **On presentation:** We rewrote the "Black-box hallucination detectors" and "Evaluation metrics" paragraphs on page 4. We fixed the Figure 1 caption typos. "affeted" becomes "affected" and "Pipeine" becomes "Pipeline".
>
> We hope the precise restatement of the claim and the explicit dissociation framing resolve your concern.